# MetaDesigner Advancing Artistic Typography through AI-Driven, User-Centric, and Multilingual WordArt Synthesis

**Jun-Yan He**[1], **Zhi-Qi Cheng**[2,3,*] **Chenyang Li**[1], **Jingdong Sun**[2], **Qi He**[2],
**Wangmeng Xiang**[1], **Hanyuan Chen**[1], **Jin-Peng Lan**[1], **Xianhui Lin**[1], **Kang Zhu**[1],
**Bin Luo**[1], **Yifeng Geng**[1], **Xuansong Xie**[1], **Alexander G. Hauptmann**[2]

[1]*Alibaba Group, Institute for Intelligent Computing (Tongyi Lab)*
[2]*Carnegie Mellon University, Language Technologies Institute (LTI)*
[3]*University of Washington, Tacoma School of Engineering & Technology (SET)*

**Project:** https://modelscope.cn/studios/WordArt/WordArt

## Abstract

*MetaDesigner* introduces a transformative framework for artistic typography synthesis, powered by Large Language Models (LLMs) and grounded in a user-centric design paradigm. Its foundation is a *multi-agent system* comprising the *Pipeline*, *Glyph*, and *Texture* agents, which collectively orchestrate the creation of customizable WordArt, ranging from semantic enhancements to intricate textural elements. A central *feedback mechanism* leverages insights from both multimodal models and user evaluations, enabling iterative refinement of design parameters. Through this iterative process, *MetaDesigner* dynamically adjusts hyperparameters to align with user-defined stylistic and thematic preferences, consistently delivering WordArt that excels in visual quality and contextual resonance. Empirical evaluations underscore the system's versatility and effectiveness across diverse WordArt applications, yielding outputs that are both aesthetically compelling and context-sensitive.

## 1 Introduction

Typography, as a nexus of linguistic expression and visual design, occupies a pivotal role across diverse fields such as advertising Cheng et al. (2016; 2017a;b); Sun et al. (2018), education Vungthong et al. (2017), and tourism Amar et al. (2017). By functioning as both a communication medium and a form of artistic expression, typography demands a nuanced understanding of aesthetics and design principles. The challenge for non-professionals lies in navigating the multifaceted considerations—ranging from visual composition to emotional resonance—necessary to produce designs that are simultaneously informative and striking.

In recent years, generative models have accelerated advancements in typographic design by enabling rapid adaptation to varied aesthetic preferences. Nevertheless, integrating these models to satisfy elaborate typesetting requirements remains non-trivial. Two significant obstacles are noteworthy: (1) *The subjective nature of artistic typography*, which varies widely based on personal and cultural contexts and complicates the development of models capable of broad appeal; and (2) *The lack of comprehensive*, annotated datasets of artistic typography, which restricts the capacity of generative models to accurately reflect the vast stylistic diversity demanded by real-world applications. While text-to-image synthesis models—such as denoising diffusion probabilistic models Ho et al. (2020); Ramesh et al. (2021); Song et al. (2021)—have made notable progress, they often struggle to address these intricacies in typography.

To tackle these challenges, we present *MetaDesigner*, a multi-agent system purpose-built to generate artistic typography guided by user preferences. As depicted in Figure 1, MetaDesigner comprises four specialized intelligent agents: the *Pipeline Designer*, *Glyph Designer*, *Texture Designer*, and *Q&A*

---

*Corresponding author. Part of this work was completed while serving as a Project Scientist at CMU.

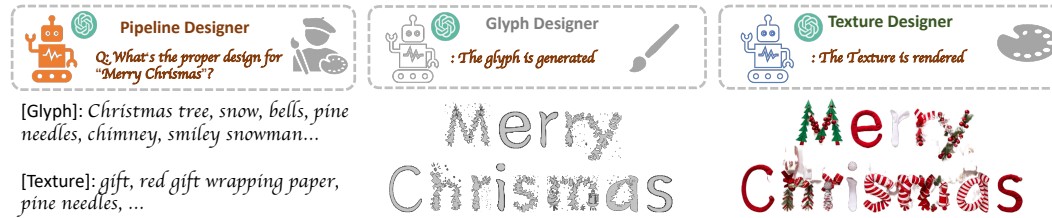

[Glyph]: *Christmas tree, snow, bells, pine needles, chimney, smiley snowman...*

[Texture]: *gift, red gift wrapping paper, pine needles, ...*

Figure 1: Overview of MetaDesigner, illustrating the interactions among the *Pipeline*, *Glyph*, and *Texture* agents, collectively shaping WordArt to align with user preferences.

*Evaluation Agent*, each contributing to personalized WordArt creation within an integrated, user-focused workflow (see Sec. 3). The *Pipeline Designer* acts as the system's orchestrator, converting user prompts into well-defined tasks for the other agents (see Sec. 3.1). The *Glyph Designer* handles diverse glyph styles—ranging from conventional lettering to semantic glyph adaptations—tailored to the thematic context of each design (see Sec. 3.2). Building on these glyphs, the *Texture Designer* applies various visual styles through LoRA model matching (see Sec. 3.3) based on a hierarchical tree structure. Finally, the *Q&A Evaluation Agent* refines the generated WordArt by integrating user feedback into an iterative, question-and-answer refinement loop (see Sec. 3.4).

The contributions of MetaDesigner are threefold:

- **Multi-agent system with evaluation and optimization.** MetaDesigner combines distinct design modules with an evaluative feedback process to discover and realize customized artistic typography styles. Through rigorous hyperparameter tuning, the platform consistently produces outputs that align with individual aesthetic preferences. Its accessibility is further evidenced by over 1,007,293 visits on ModelScope[1].

- **Advanced glyph design through a hierarchical model tree.** The *Glyph Designer* leverages both comprehensive font libraries and semantic translation methods to support extensive glyph transformations. Equipped with a hierarchical tree of 68 LoRA models, MetaDesigner ensures rich stylistic diversity, significantly broadening the creative possibilities in typographic design.

- **Comprehensive dataset fostering continued research.** To further support the study of artistic typography, we introduce a carefully curated dataset containing over 5,000 multilingual images (English, Chinese, Japanese, and Korean) spanning an array of artistic styles and cultural elements. This resource provides researchers and practitioners with a robust foundation for benchmarking new techniques and advancing the state of the art in artistic text generation.

## 2 RELATED WORK

**Text-to-Image Synthesis.** Recent advancements in denoising diffusion probabilistic models Ho et al. (2020); Song et al. (2021); Dhariwal & Nichol (2021); Nichol & Dhariwal (2021); Saharia et al. (2022); Ramesh et al. (2022); Rombach et al. (2022); Chang et al. (2023) have significantly enhanced the fidelity and flexibility of text-to-image synthesis. These models have given rise to interactive editing approaches Meng et al. (2022); Gal et al. (2023); Brooks et al. (2022) and multi-condition controllable pipelines Zhang & Agrawala (2023); Mou et al. (2023); Huang et al. (2023). Recent work such as ELITE Wei et al. (2023), UMM-Diffusion Ma et al. (2023b), and InstantBooth Shi et al. (2023) leverages CLIP-based image encoders to bridge visual information with textual embeddings, expanding the expressive range of generative models.

**Visual Text Generation.** Producing legible text within images poses substantial challenges Rombach et al. (2022), especially when dealing with complex or multilingual scripts. Approaches like *GlyphDraw* Ma et al. (2023a) and *GlyphControl* Yang et al. (2023) address character alignment and rendering, while *TextDiffuser* Chen et al. (2023b) introduces character-level segmentation to enhance text clarity. Large-scale language models further refine text generation Saharia et al. (2022); Balaji et al. (2022); Lab (2023), yet traditional text encoders often struggle with non-Latin alphabets Liu et al. (2023). To address such issues, *GlyphDraw* fine-tunes text encoders and integrates CLIP-based glyph embeddings Ma et al. (2023a), whereas *DiffUTE* uses pre-trained image encoders to extract glyphs for editing tasks Chen et al. (2023a).

---

[1]Project: https://modelscope.cn/studios/WordArt/WordArt

**WordArt Synthesis.** The synthesis of WordArt Tanveer et al. (2023); Iluz et al. (2023); Berio et al. (2022); Tendulkar et al.; Zhang et al. (2017) aims to integrate semantic richness with artistic yet readable text representations. Early work Tendulkar et al. experimented with replacing letters by semantically related icons. More recent models harness large-scale generative techniques to push typographic boundaries. For example, *Word-As-Image* Iluz et al. (2023) proposes artistic typography for Latin scripts, while *DS-Fusion* Tanveer et al. (2023) explores the generation of intricately styled texts, including hieroglyphs.

**Positioning of MetaDesigner.** Unlike prior methods, *MetaDesigner* adopts a multi-agent framework featuring *Pipeline*, *Glyph*, *Texture*, and *Q&A Evaluation* agents within an interactive feedback loop. This architecture iteratively refines semantic, glyph, and texture elements based on user input, covering a broader range of aesthetic demands and typographic nuances than existing approaches.

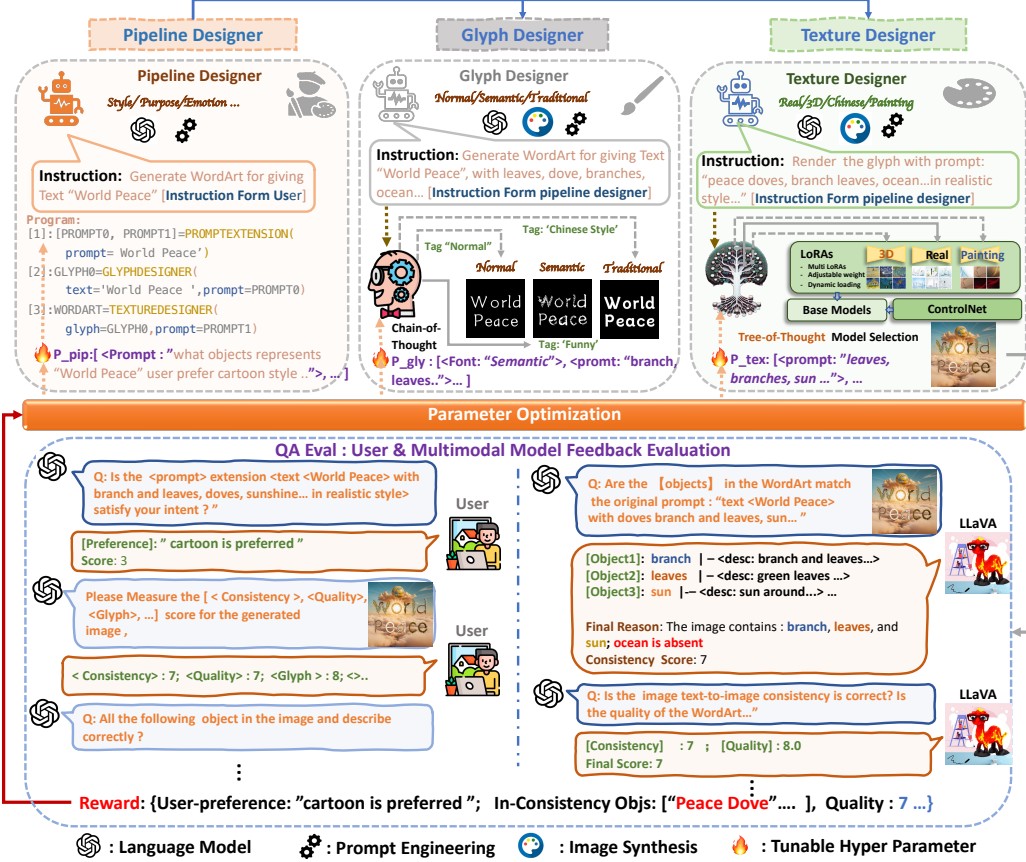

Figure 2: **MetaDesigner Architectural Overview.** The framework integrates three primary intelligent agents—*Pipeline Designer*, *Glyph Designer*, and *Texture Designer*—to produce personalized WordArt. A *Q&A Evaluation* agent runs in parallel to iteratively refine the design. This diagram highlights how textual inputs are transformed into visually compelling, user-driven artistic typography through an interactive, iterative process.

## 3 METADESIGNER FRAMEWORK

**Overview.** *MetaDesigner* is an interactive multi-agent system for synthesizing user-tailored WordArt. Four specialized agents—*Pipeline Designer*, *Glyph Designer*, *Texture Designer*, and *Q&A Evaluation Agent*—collaborate to generate and refine typographic art according to evolving user preferences.

**Mathematical Formulation.** Let $\hat{I}$ be the synthesized WordArt, defined by an operator $\Psi$:

$$\hat{I} = \Psi\big(s^{\text{user}}, \phi, \mathcal{P}, \mathcal{M}\big), \tag{1}$$

where $s^{\text{user}}$ is the user prompt (specifying style, theme, or other attributes), $\phi = \{\phi^{\text{pip}}, \phi^{\text{gly}}, \phi^{\text{tex}}, \phi^{\text{qa}}\}$ encompasses the functionalities of the four agents, $\mathcal{M}$ denotes a model library (used primarily by the *Texture Designer*), and $\mathcal{P} = \{\mathcal{P}^{\text{pip}}, \mathcal{P}^{\text{gly}}, \mathcal{P}^{\text{tex}}, \mathcal{P}^{\text{qa}}\}$ represents hyperparameters tunable via feedback.

**Iterative Process.** To adapt and improve results across iterations, MetaDesigner leverages previously generated images. Let $I_{\text{prev}}$ be the output of a past iteration. The synthesis process then becomes:

$$
\begin{aligned}
\hat{I} &= \Psi\big(s^{\text{user}}, \phi, \mathcal{P}, \mathcal{M}\big) \\
&= \phi^{\text{pip}}\big(s^{\text{user}}\big) \ \cdot \ \phi^{\text{tex}}\big(\phi^{\text{gly}}(s^{\text{gly}}), \, s^{\text{tex}}, \, \mathcal{M}\big) \ \cdot \ \phi^{\text{qa}}\big(I_{\text{prev}}, \mathcal{P}^{\text{qa}}\big),
\end{aligned}
\tag{2}
$$

where $\phi^{\text{pip}}$ produces an extended prompt $\mathbf{S}$ from $s^{\text{user}}$. This extended prompt divides into $\{s^{\text{gly}}, s^{\text{tex}}\}$, guiding the *Glyph* and *Texture* agents.

**Feedback and Adaptation.** The *Q&A Evaluation Agent* reviews both the previously generated image $I_{\text{prev}}$ and hyperparameters $\mathcal{P}^{\text{qa}}$, then incorporates user and automated feedback to refine system parameters. Through this loop, MetaDesigner incrementally aligns outputs with user preferences, producing visually coherent and context-aware WordArt. As shown in Figure 2, the sections below detail the functionalities of each agent, illustrating their collaborative roles in the synthesis pipeline.

## 3.1 PIPELINE DESIGNER AGENT

The *Pipeline Designer* is responsible for translating user instructions into a structured workflow, ensuring seamless coordination among the remaining agents. By combining **visual programming** techniques with a **Chain-of-Thought (CoT)** prompting strategy, this component simplifies the design process into manageable steps while incorporating iterative feedback to refine the final output.

**Visual Programming.** As depicted in Figure 2, GPT-4's in-context learning capabilities are employed to generate visual programs directly from natural language instructions. These programs, assembled without explicit fine-tuning, consist of sequential "blocks" or "modules," each defined by:

- A *module name*, identifying the function or operation.
- A set of *input arguments* with specified names and values.
- An *output variable name* that receives the module's result.

This modular structure enforces clarity in the synthesis pipeline, allowing straightforward validation and debugging of intermediate steps.

**Prompt Extension.** To tailor output to user preferences, the *Pipeline Designer* employs CoT reasoning within GPT-4. Beginning with a user prompt $s^{\text{user}}$, the system generates an enriched prompt $\mathbf{S}$ that includes specialized instructions for the *Glyph Designer* and *Texture Designer*:

$$
\mathbf{S} = \phi^{\text{pip}}(s^{\text{user}}) = \{\, s^{\text{gly}}, \, s^{\text{tex}}\}.
\tag{3}
$$

Here, $\phi^{\text{pip}}$ represents the *Pipeline Designer* function. By posing clarifying questions related to style, application context, and design constraints, GPT-4 refines the initial prompt into two well-defined components: $s^{\text{gly}}$ (for glyph design) and $s^{\text{tex}}$ (for texture design).

**Feedback Integration.** A key advantage of the *Pipeline Designer* lies in its ability to incorporate user feedback in an iterative loop. Formally, feedback is modeled via a function $\mathcal{F}$ that aggregates various signals $G$ (e.g., user evaluations, automated metrics), along with an update directive $s^{\text{update}}$:

$$
\mathcal{P}^{\text{pip}}_{\text{new}} = \mathcal{F}\big(G \mid s^{\text{update}}\big),
\tag{4}
$$

where $\mathcal{P}^{\text{pip}}_{\text{new}}$ denotes the updated set of hyperparameters for the pipeline. The *Pipeline Designer* also returns the enriched prompt $\mathbf{S}$ needed by downstream agents:

$$
\mathbf{S}, \mathcal{P}^{\text{pip}}_{\text{new}} \ = \ \phi^{\text{pip}}\big(s^{\text{user}}, \mathcal{P}^{\text{pip}}, \mathcal{F}(G \mid s^{\text{update}})\big).
\tag{5}
$$

This guarantees that subsequent design iterations progressively align more closely with user intent. Through active integration of user preferences and continuous prompt refinement, the *Pipeline Designer* ensures that each step of the workflow converges toward the most suitable artistic outcome.

## 3.2 GLYPH DESIGNER AGENT

The *Glyph Designer* is central to MetaDesigner's pipeline, capable of producing three distinct glyph types—*Normal*, *Traditional*, and *Semantic*—to accommodate a wide spectrum of stylistic needs. While *Normal* and *Traditional* glyphs are suited for formal contexts (e.g., weddings, galas), *Semantic* glyph transformations cater to more imaginative or humorous applications. GPT-4 automatically determines the most appropriate glyph type based on the user prompt.

**Normal & Traditional Glyph Generation.**   In formal scenarios, the system renders conventional or culturally traditional glyphs with the FreeType font library:

$$G_n, \, G_t = \phi_n(s^{\text{gly}}), \quad \phi_t(s^{\text{gly}}) \tag{6}$$

where $\phi_n$ and $\phi_t$ are specialized rendering functions, and $s^{\text{gly}}$ denotes the glyph-specific prompt. This setup ensures a clean, dignified aesthetic for occasions demanding a formal tone.

**Semantic Glyph Transformation.**   For creative and humorous contexts, the Glyph Designer supports *semantic glyph transformations* through $\phi_s$. These transformations reshape vector-based glyphs to approximate target objects or thematic concepts, leveraging differentiable rasterization and a depth-to-image Stable Diffusion model:

$$G_s = \phi_s(s^{\text{sem}}, \mathcal{M}), \tag{7}$$

where $s^{\text{sem}}$ is a semantic prompt capturing the desired concept, and $\mathcal{M}$ references the model library. During this process, vector parameters in an SVG representation are iteratively optimized so that the resulting glyph visually integrates the target idea without compromising legibility.

**Optimization Mechanism.**   To achieve smooth, context-aware deformations, the system begins by creating a glyph image $I_{\text{sem}}$ from trainable parameters $\theta$ using DiffVG. The character segment $x$ is then cropped and augmented into batches $X_{\text{aug}}$. Coupled with a semantic concept $S$, these batches are fed into a vision-language model to calculate the training loss—particularly the SDS loss in the latent space code $z$. This iterative procedure refines $\theta$, producing a final glyph design that balances aesthetic appeal with textual clarity.

**Glyph Style Selection.**   Finally, the glyph style is determined by context-derived cues, enabling both aesthetic and functional considerations. Formally:

$$G = \phi\big(s^{\text{gly}}, \mathcal{P}^{\text{gly}}, \mathcal{M}\big) = \begin{cases} \phi_n(s^{\text{gly}}), & \text{if formal context,} \\ \phi_t(s^{\text{gly}}), & \text{if traditional context,} \\ \phi_s(s^{\text{sem}}, \mathcal{M}), & \text{if creative context.} \end{cases} \tag{8}$$

Where $G$ is the final glyph, and $\mathcal{P}^{\text{gly}}$ represents glyph-specific hyperparameters. By choosing the appropriate rendering technique and iteratively refining vector shapes, the *Glyph Designer* produces glyphs that are visually appealing while fulfilling functional needs.

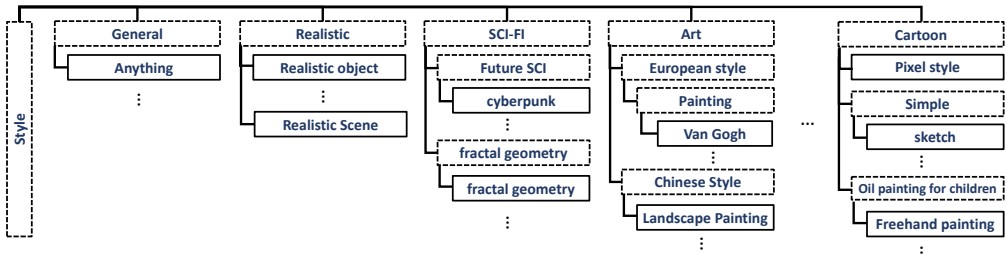

Figure 3: A hierarchical model tree with multiple sub-categories for fine-grained **ToT** model selection.

## 3.3 TEXTURE DESIGNER AGENT

The *Texture Designer* enriches glyphs by integrating controllable image synthesis with a *Tree-of-Thought (ToT)* strategy, ensuring both creative diversity and alignment with user preferences. Formally, it can be represented as follows:

$$I_{\text{tex}} = \psi^{\text{tex}}\big(I_{\text{gly}},\, s^{\text{tex}},\, \mathcal{F},\, \mathcal{M}\big)$$
$$= \mathbf{TexR}\Big(I_{\text{gly}},\, s^{\text{tex}},\, C,\, \mathbf{ToTSel}\big(s^{\text{tex}},\, \mathcal{F},\, \mathcal{M}\big)\Big), \tag{9}$$

where $I_{\text{tex}}$ is the final textured output, **TexR** denotes the controllable synthesis mechanism, **ToTSel** encapsulates the ToT-based model selection function, $I_{\text{gly}}$ is the input glyph image, $s^{\text{tex}}$ is the texture prompt, $C$ represents control conditions (e.g., edge maps, depth), $\mathcal{F}$ guides the ToT process, and $\mathcal{M}$ is the model library. The remainder of this subsection provides a high-level overview; additional details can be found in the supplementary materials.

**Controllable Synthesis.** Building on ControlNet, the Texture Designer manipulates stylization parameters such as edges, depth, or scribbles to realize varied and adaptive texture styles:

$$I_{\text{tex}} = \mathbf{TexR}\big(I_{\text{gly}},\, s^{\text{tex}},\, C,\, \mathcal{W}\big), \tag{10}$$

where $\mathcal{W}$ denotes the selected model weights. By tweaking these parameters, the system can fine-tune visual details, catering to a range of aesthetic or thematic requirements.

**Tree-of-Thought Selection.** To ensure artistic originality and coherence with user preferences, the Texture Designer adopts a Tree-of-Thought approach (see Figure 3). First, the prompt $s^{\text{tex}}$ is decomposed into conceptual pathways $\{z_1, z_2, \dots, z_n\}$ under the guidance of $\mathcal{F}(s^{\text{tex}})$. Each pathway's suitability is then evaluated by a heuristic $V(z_i)$, leading to a search for the model $\mathcal{M}_{\text{best}}$ that maximizes the aggregate score:

$$\mathcal{M}_{\text{best}} = \underset{\mathcal{M} \in \mathcal{M}_{\text{lib}}}{\arg\max} \sum_{i=1}^{n} V\big(z_i \mid \mathcal{M}\big). \tag{11}$$

This approach identifies the model that best matches each conceptual path, thereby enhancing the creative alignment between the chosen style and the user's vision.

**Model Library Integration.** Within MetaDesigner, a curated library of 68 LoRA models (see Figure 11) spans categories like "General," "Realistic," "SCI-FI," "Art," "Design," and "Cartoon." To maximize flexibility, these models can be fused through weighted combinations:

$$\mathcal{W}_{\text{fusion}} = \sum_{i} \alpha_i\, \mathcal{W}_i, \tag{12}$$

where $\alpha_i$ denotes the blending coefficient for each LoRA model $\mathcal{W}_i$. By adapting the Tree-of-Thought output to this model library, the Texture Designer deftly transforms plain glyph images into textural "masterpieces," balancing user-defined constraints with artistic flair.

### 3.4 Q&A Evaluation Agent

MetaDesigner incorporates a feedback mechanism designed to fine-tune hyperparameters based on four core criteria: **text-to-image consistency**, **image quality**, **glyph feedback**, and **user preference**. The system employs the LLaVA model to quantitatively assess consistency and overall image quality, while user studies contribute qualitative feedback via a Q&A format.

**Feedback Metrics.** For each newly synthesized image, GPT-4 generates evaluation prompts that encapsulate the project's goals (e.g., artistic style, thematic adherence). LLaVA analyzes these prompts against the generated images and outputs a metric set:

$$G_m = \{\, g_m^{\text{cos}},\, g_m^{\text{qua}},\, \mathcal{L}_m \,\},$$

where $g_m^{\text{cos}}$ denotes text-to-image consistency, $g_m^{\text{qua}}$ represents image quality, and $\mathcal{L}_m$ is a loss-like term used in optimizing system parameters. GPT-4 then summarizes these findings, producing coherent feedback and suggesting rationales for potential hyperparameter updates.

**User Feedback.** In parallel, MetaDesigner gathers optional user responses through GPT-4-initiated Q&A sessions, capturing preferences ($g_u^{\text{pref}}$) and perceptions of glyph style ($g_u^{\text{gly}}$). Formally:

$$G_u = \{ g_u^{\text{cos}}, g_u^{\text{qua}}, g_u^{\text{gly}}, g_u^{\text{pref}}, \mathcal{L}_u \}.$$

Although not mandatory, these user evaluations provide crucial insight into subjective factors like aesthetic appeal or contextual appropriateness. MetaDesigner integrates all feedback via

$$G = \text{Merge}\big(G_m, \, G_u\big),$$

giving precedence to user input if any conflicts arise.

**Optimization Strategy.** Drawing on both model-based and user-derived inputs, MetaDesigner frames the overall optimization objective as $\mathcal{L} = \mathcal{L}_m + \mathcal{L}_u$. Specifically,

$$\mathcal{L}_m = \underset{\{\mathcal{P}^{\text{gly}}, \mathcal{P}^{\text{tex}}\}}{\text{argmax}} \ \mathcal{H}\big(s^{\text{eval}}, \hat{I}\big), \tag{13}$$

where $s^{\text{eval}}$ is the evaluative prompt and $\mathcal{H}$ the LLaVA-based heuristic measuring how well the generated image $\hat{I}$ satisfies the evaluation criteria. Accordingly, the system adjusts glyph design parameters $\mathcal{P}^{\text{gly}}$ and texture design parameters $\mathcal{P}^{\text{tex}}$ to maximize this composite objective.

**Iterative Hyperparameter Tuning.** As illustrated in Algorithm 1 and Figure 4, the system dynamically refines its entire hyperparameter set $\mathcal{P} = \{\mathcal{P}^{\text{pip}}, \mathcal{P}^{\text{gly}}, \mathcal{P}^{\text{tex}}, \mathcal{P}^{\text{qa}}\}$ by applying the feedback function $\hat{\mathcal{P}} = \mathcal{F}\big(G \mid s^{\text{update}}\big)$. Generally, user preferences and glyph-specific cues influence $\mathcal{P}^{\text{gly}}$ and pipeline adjustments, while text-to-image alignment and quality assessments inform $\mathcal{P}^{\text{tex}}$. This closed-loop process ensures continual convergence toward outcomes that balance objective visual performance with the subjective nuances of user-defined style.

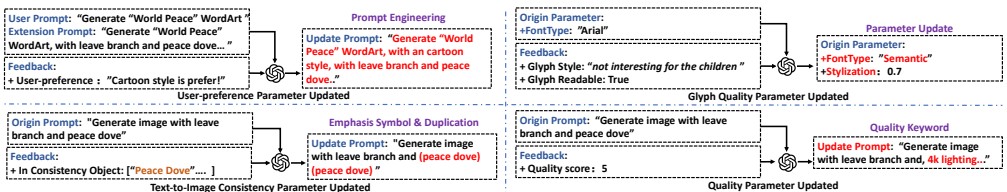

Figure 4: **Feedback Loop for Hyperparameter Tuning:** An overview of how user preferences, glyph quality, text-to-image consistency, and image quality assessments iteratively guide hyperparameter updates.

## 4 EXPERIMENTS

To evaluate the effectiveness of *MetaDesigner*, we curated 150 prompts spanning five themes—*cartoon*, *design*, *reality*, *sci-fi*, and *traditional culture*—encompassing a wide range of artistic styles and cultural motifs. From these, we selected 20 prompts (in English, Chinese, Japanese, and Korean) for user studies to investigate the framework's multilingual performance.

### 4.1 COMPARISON WITH STATE-OF-THE-ART METHODS

We compared *MetaDesigner* against several contemporary state-of-the-art (SOTA) models, including Stable Diffusion XL (SD-XL) Podell et al. (2023), TextDiffuser Frans et al. (2022), TextDiffuser-2 Chen et al. (2023c), Anytext Tuo et al. (2023), and DALL-E 3. The models were chosen to represent a spectrum of approaches in text-to-image synthesis. Representative results are illustrated in Figure 5 and evaluated on the following criteria:

**WordArt Synthesis Success.** Figure 5 shows that SD-XL often fails to accurately depict text, performing inconsistently even in English. Both versions of TextDiffuser produce acceptable English WordArt but encounter difficulties with Chinese, Korean, and Japanese. Anytext improves performance on English yet struggles with Korean and Japanese, and DALL-E 3 is similarly restricted to English. By contrast, *MetaDesigner* adeptly handles all tested languages, demonstrating robust, high-quality WordArt generation. (See supplementary materials for additional examples.)

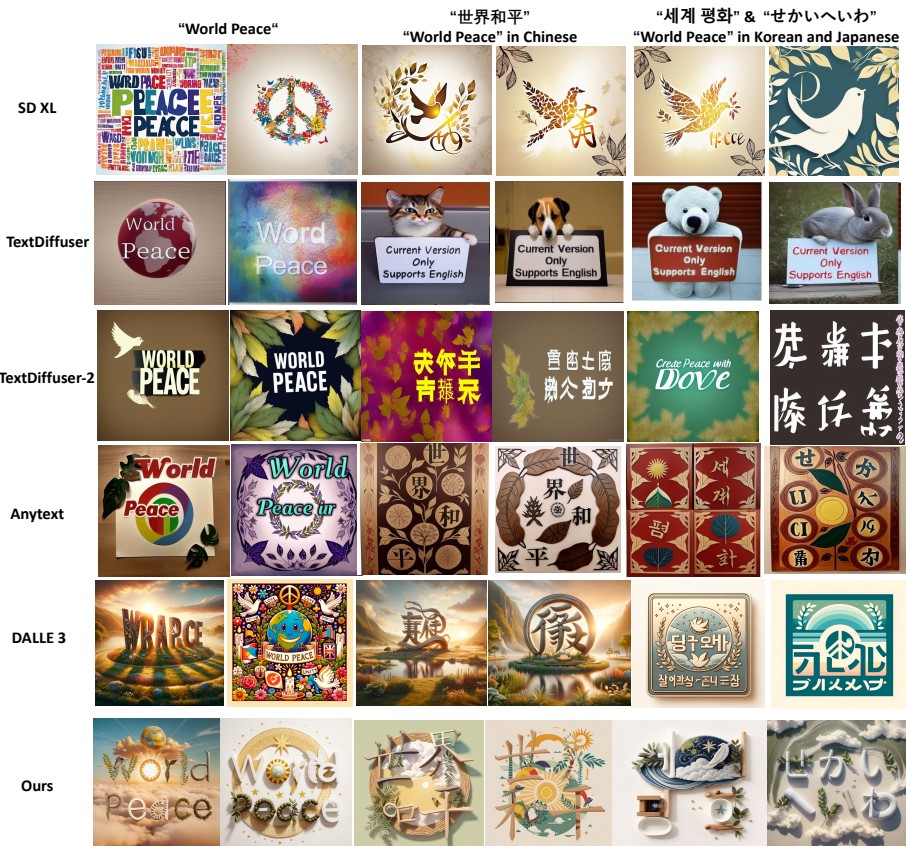

Figure 5: **WordArt Synthesis Comparison:** Columns 1 and 2 illustrate "World Peace" in English, columns 3 and 4 present the Chinese rendition, and columns 5 and 6 show the Korean and Japanese versions, respectively. The leftmost column corresponds to the baseline prompt ("Create a stylish word 'World Peace' representing its meaning"), while subsequent columns include additional keywords such as "Sun, Peace Dove, leaves, cloud."

**Quality and Diversity.** Owing to its reliance on the Stable Diffusion 1.5 backbone and limited training data, the TextDiffuser family tends to yield visually similar results across prompts. While Anytext exhibits greater stylistic variety, it occasionally sacrifices image fidelity. DALL-E 3 achieves higher-quality outputs but tends to default to a cinematic or 3D aesthetic. As shown in Figure 5, *MetaDesigner* excels in both quality and stylistic breadth, consistently producing diverse outputs—ranging from realistic to cartoon or 3D—while preserving overall visual quality.

**Creativity and Semantic Alignment.** Although SD-XL produces visually appealing text, it often strays from the prompt's meaning. The TextDiffuser series includes thematic elements (e.g., leaves and a peace dove for "World Peace") but lacks cultural or linguistic adaptability. Anytext sometimes introduces unrelated logos that miss the prompt's essence, and DALL-E 3, while visually refined, can be thematically ambiguous. In contrast, *MetaDesigner* consistently generates creative, semantically coherent WordArt (e.g., adding a dove and foliage for "World Peace"), demonstrating a clear grasp of both context and theme (see Figure 5 and Figure 9).

**Quantitative Analysis.** Because OCR methods often struggle with stylized text, we conducted a user study to evaluate *MetaDesigner* alongside competing approaches on two dimensions: *Text Accuracy* and *Aesthetics & Creativity*. Eleven participants provided feedback, ensuring varied perspectives. As seen in Table 1, *MetaDesigner* clearly outperforms other methods in both criteria, striking a strong balance between textual fidelity and visual appeal. We further assessed each method using SSIM (Structural Similarity) and LPIPS (Learned Perceptual Image Patch Similarity). Table 2 shows *MetaDesigner* consistently achieving superior results on both metrics, underscoring its capacity to produce readable, visually coherent WordArt.

**Letter-Level Comparison.** We further evaluated *MetaDesigner* at a more granular level by comparing its single-letter WordArt outputs against Google search results, Stable Diffusion Rombach et al.

Table 1: **User Study Results.** Higher scores are better; * denotes *MetaDesigner* without feedback.

| Dimension | SDXL | SDXL-Aug | TextDiff | TextDiff-2 | Anytext | DALLE 3 | Ours* | Ours |
|---|---|---|---|---|---|---|---|---|
| **Text Accuracy** | 3.6 | 3.6 | 44.1 | 42.7 | 82.3 | 33.2 | 88.2 | **96.8** |
| **Aesthetics** | 0.5 | 5.9 | 0.9 | 0.5 | 0.9 | 10.9 | 5.0 | **75.4** |
| **Creativity** | 0.9 | 1.8 | 0.5 | 0.9 | 0.9 | 7.3 | 10.5 | **77.2** |

Table 2: **Quantitative Comparison.** We measure SSIM and LPIPS on ground-truths from Promeai ($\mathcal{P}$) and design websites ($\mathcal{D}$). **Bold** and underlined are best and second-best. * denotes *MetaDesigner* w/o feedback.

| GT | Metric | Text | SDXL | SDXL-Aug. | TDiff | TDiff-2 | Anytext | DALLE 3 | Ours* | Ours |
|---|---|---|---|---|---|---|---|---|---|---|
| $\mathcal{P}$ | SSIM↑ | Eng. | 0.1254 | 0.1381 | 0.1860 | 0.1641 | 0.1324 | 0.0834 | 0.1730 | **0.2397** |
| | | CJK | 0.1853 | 0.2092 | 0.1747 | 0.2037 | 0.1021 | 0.1401 | 0.2269 | **0.2643** |
| | | All | 0.1553 | 0.1736 | 0.1803 | 0.1839 | 0.1172 | 0.1117 | 0.2000 | **0.2520** |
| | LPIPS↓ | Eng. | 0.7491 | 0.7684 | 0.7652 | 0.7441 | 0.7453 | 0.7653 | 0.6960 | **0.6910** |
| | | CJK | 0.7712 | 0.7307 | 0.7970 | 0.7687 | 0.7601 | 0.7693 | 0.6937 | **0.6846** |
| | | All | 0.7602 | 0.7496 | 0.7811 | 0.7564 | 0.7527 | 0.7673 | 0.6949 | **0.6878** |
| $\mathcal{D}$ | SSIM↑ | Eng. | 0.1802 | 0.2439 | 0.2342 | 0.2036 | 0.1669 | 0.1413 | 0.1913 | **0.3119** |
| | | CJK | 0.1951 | 0.2093 | 0.1846 | 0.1987 | 0.1073 | 0.1542 | 0.2184 | **0.2539** |
| | | All | 0.1877 | 0.2266 | 0.2094 | 0.2012 | 0.1371 | 0.1478 | 0.2048 | **0.2829** |
| | LPIPS↓ | Eng. | 0.7993 | 0.8157 | 0.8312 | 0.8366 | 0.8495 | **0.7650** | 0.8169 | 0.7950 |
| | | CJK | 0.8023 | 0.7964 | 0.8249 | 0.8429 | 0.8437 | **0.7872** | 0.7912 | 0.7880 |
| | | All | 0.8008 | 0.8060 | 0.8280 | 0.8397 | 0.8466 | **0.7761** | 0.8040 | 0.7915 |

(2022), DALL-E 3, and DS-Fusion Tanveer et al. (2023). As shown in Figure 8, Stable Diffusion struggles with coherent letter shapes, while DS-Fusion produces cleaner forms but offers limited style variation. DALL-E 3 demonstrates high textural quality yet does not always capture thematic nuances. In contrast, *MetaDesigner* consistently merges artistic creativity with precise detailing, achieving visually appealing, thematically aligned letters. More examples are in the supplementary material.

## 4.2 EFFECT OF TREE-OF-THOUGHT

To verify the effectiveness of the Tree-of-Thought (ToT) scheme, we conducted a quantitative analysis comparing the ControlNet and the ToT-LoRA+ControlNet approaches. GPT-4, a state-of-the-art language model, was used to measure the "Relevance," "Quality," and "Style" of the synthesized WordArt. The results, illustrated in Figure 6, show significant improvements achieved by the ToT-LoRA in all metrics, highlighting its ability to generate WordArt that is highly relevant to the given prompts, visually appealing, and stylistically diverse.



Figure 6: The evaluation of the synthesis WordArt. From left to right are the "Relevance", "Quality", and "Style" scores generated by the ChatGPT-4 in the subcategories. (The performance of ToT+Gemini and ToT+GPT4 are very close)

A case study, presented in Figure 7, demonstrates that the ToT-LoRA scheme significantly outperforms ControlNet in WordArt style and text-to-image consistency. Specifically, ToT-LoRA+ControlNet excels in rendering complex textures and maintaining coherence with the given prompts, such as "ethnic customs, New Year, dumplings, steamed bread, kitchen, mother, little boy" and "Renaissance style, Cas-

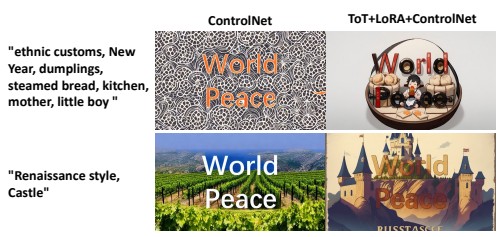

Figure 7: WordArt Texture rendering.

tle." The results show clearer thematic representation and enhanced visual appeal, further validating the effectiveness of our approach. More examples are in the supplementary material.

## 4.3 EFFECT OF OPTIMIZATION

To illustrate the impact of the optimization process, we conducted a detailed case study, presented in Figure 10. The LLaVA system is employed to identify objects mentioned in the prompt but absent in the generated WordArt. This information is then used to update the generation process, incorporating the omitted elements. Techniques such as symbol enhancement, word sequencing adjustments, and

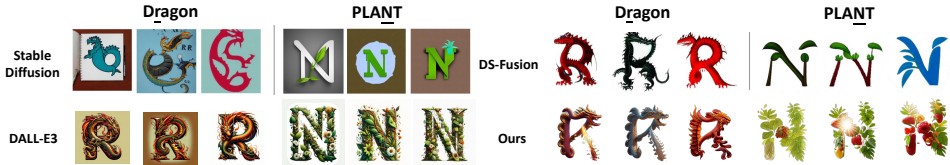

Figure 8: **Single-Letter WordArt Comparison:** We compare Google, Stable Diffusion, DALL-E 3, DS-Fusion, and *MetaDesigner*. The prompts here are "Dragon" (left) and "Plant" (right). Partial results from DS-Fusion Tanveer et al. (2023) are reproduced for reference.

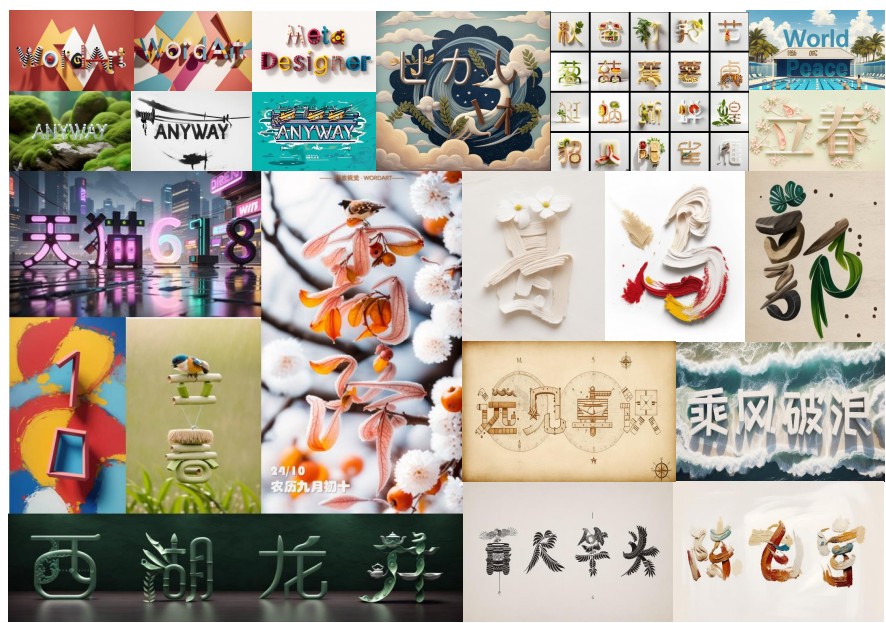

Figure 9: Examples showcasing multi-language support (Chinese, English, Japanese, Korean, Arabic numerals, etc.) and varied word counts, with applications ranging from e-commerce to personalized social media.

keyword repetition are employed to augment the WordArt generation process, ensuring that the final output accurately reflects the input prompt. As shown in Figure 10, the optimization process proceeds through several steps to achieve better alignment with the prompts. In Step 1, the initial generation may miss key elements such as "little girl" in the prompt "old man, cake, candles, little girl." In Step 2, iterative refinement introduces missing elements, enhancing text-to-image consistency. Similarly, for the prompt "kitchen, girl, steamed bread, a plate of fruit," the optimization process adds missing objects and refines their depiction over multiple steps. This optimization process plays a crucial role in enhancing the semantic consistency and visual quality of the generated WordArt.

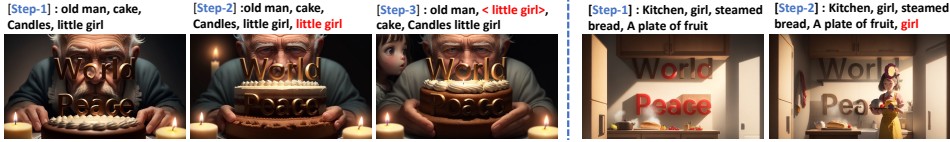

Figure 10: The examples are the optimization of the text-to-image consistency.

## 5 CONCLUSION

We introduced *MetaDesigner*, an LLM-driven, multi-agent framework that streamlines user-centric artistic typography synthesis. Its three core agents—*Pipeline*, *Glyph*, and *Texture*—collaboratively translate user preferences into visually compelling, context-aware WordArt. By integrating generative AI with typographic rendering, *MetaDesigner* enables both professionals and hobbyists to efficiently produce high-quality, customizable, and aesthetically diverse designs. Future work includes expanding the WordArt Dataset, extending language coverage, refining the multi-agent system, and exploring broader applications in design, branding, digital media, and visual communication.

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

## A  APPENDIX: ADDITIONAL DETAILS

This appendix provides further information that could not be included in the main paper. The sections are organized as follows:

1. **ToT Model Selection** (appendix B)

    (a) *ToT Model Selection Prompt* (appendix B.1)
    (b) *ToT Model Selection Case Study* (appendix B.2)

2. **Image Evaluation** (appendix C)

    (a) *ToT-LoRA Evaluation Prompt* (appendix C.1)
    (b) *LLaVA Evaluation Prompt* (appendix C.2)

3. **Dataset Details** (appendix D)
4. **Additional Comparisons with SOTA Methods** (fig. 17)
5. **Mixing vs. Separating the Glyph and Texture Designers** (fig. 18)
6. **Effect of GPT-4/GPT-4V on ToT Selection** (appendix E)

## B  MORE DETAILS OF TOT MODEL SELECTION

### B.1  TOT MODEL SELECTION TEMPLATE

*You possess knowledge of various cultural backgrounds and artistic styles. For each prompt, identify and analyze its vocabulary, theme, content, implied culture, and potential reader perception. From this analysis, choose the most appropriate element from the given input list. Document the reasoning steps, then determine a final consensus or best-guess match. The final output must adhere to the following template:*

*TEMPLATE:*

```
Selected:  [the selected word]
Input list:  {search_list}
Input prompt:  {input}
```

### B.2  TOT MODEL SELECTION CASE

*Input Prompt:* `A girl, a boy, in a church`

- **Thoughts-1st (Keywords: "General", "Traditional Art", "Design", "Cartoon", "SCI-FI", "Realistic")**

    – **Vocabulary:** The prompt uses common words (girl, boy, church) that do not point to a specific art style or culture.
    – **Theme:** Featuring a girl and a boy in a church, suggesting a spiritual or religious setting often associated with traditional art.
    – **Content:** Minimal details; however, the church context is frequently linked to traditional or classical depictions.
    – **Implicit Culture:** Likely Western or Christian influence, common in traditional art.
    – **Reader Perception:** A straightforward scene that leans toward traditional aesthetics in many cultural contexts.
    – **Summarization:** "Traditional Art" best fits this simple yet possibly religious scene.
    – **Selected:** *Traditional Art*

- **Thoughts-2nd (Keywords: "European Style", "Chinese Style")**

    – **Vocabulary:** The prompt mentions a girl, a boy, and a church, but does not specify a distinct culture or art form.
    – **Theme:** Two individuals in a religious setting, with no direct mention of cultural influence.
    – **Content:** Minimal context, though a church often implies a Western or European setting.

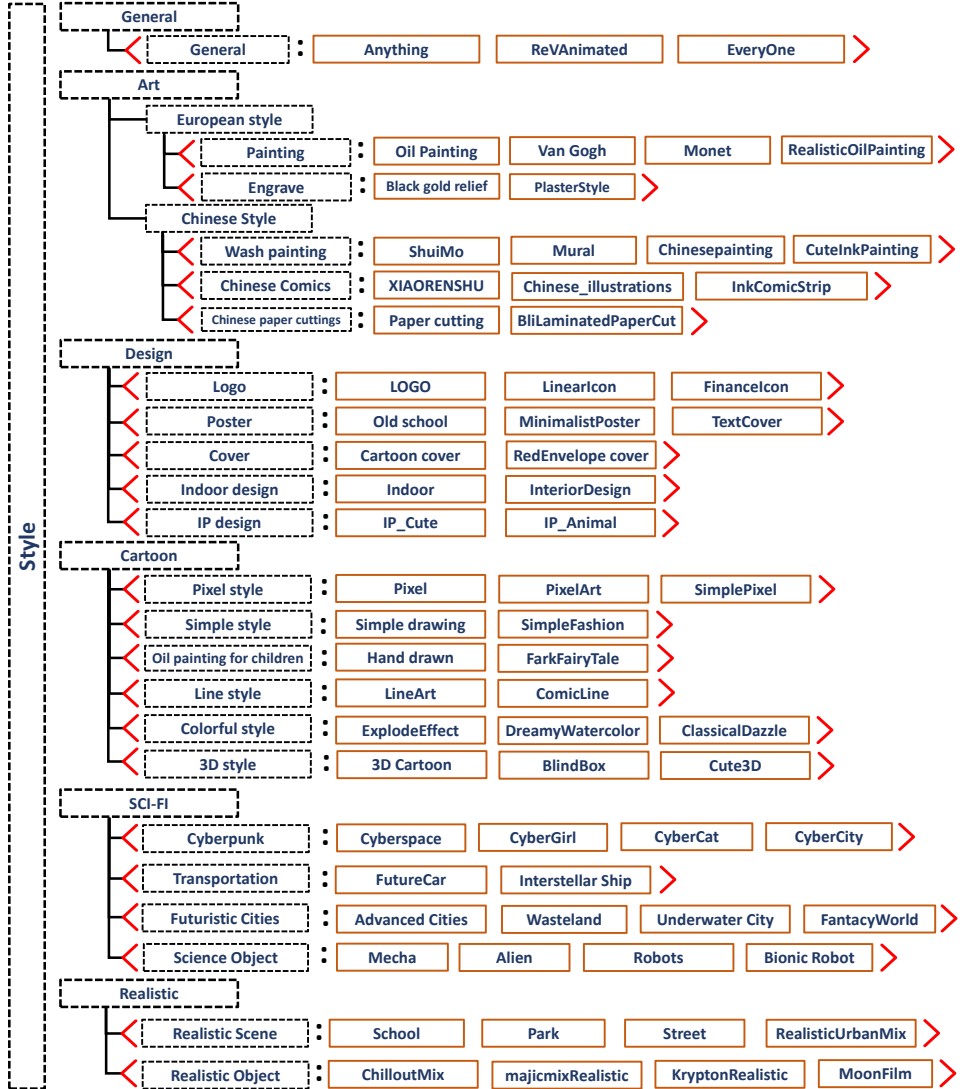

Figure 11: The complete hierarchical model tree, featuring multi-level subcategories for in-depth **ToT** model selection and enhanced stylistic diversity.

- **Implicit Culture:** Likely European, given the mention of a church.
- **Reader Perception:** Many viewers may perceive a Western influence, though this remains subjective.
- **Summarization:** "European Style" appears most compatible with the church setting.
- **Selected:** *European style*

• **Thoughts-3rd (Keywords: "Painting", "Engrave")**

- **Vocabulary:** Plain language referencing a girl, boy, and church, making the setting easily visualized.
- **Theme:** Suggests a peaceful or spiritual scenario with no explicit action, often depicted in paintings.
- **Content:** Minimalistic, focusing on two characters within a church.
- **Implicit Culture:** The mention of a church indicates a likely Christian or Western environment.
- **Reader Perception:** Evokes serenity or spirituality; could be rendered effectively as a painting.
- **Summarization:** "Painting" suits the subtle details of the scene better than "Engrave."
- **Selected:** *Painting*

---

**Algorithm 1** Hyperparameter Tuning

---

1: **Input:** Prompt $s^{\text{user}}$, initial hyperparameters $\mathcal{P}$, max iteration threshold $\tau$, score threshold $\theta$, model library $\mathcal{M}$, MetaDesigner $\Psi$;
2: **Output:** WordArt image $\hat{I}$;
3: **while** $i < \tau$ and $\mathcal{L} < \theta$ **do**
4:     $\hat{I} = \Psi(s^{\text{user}}, \phi, \mathcal{P}, \mathcal{M});$                                             ▷ Eq. (1)
5:     $G_m = \mathcal{H}(s^{\text{eval}}, \hat{I})$
6:     $G_u = \{g_u^{\text{cos}}, g_u^{\text{qua}}, g_u^{\text{tex}}, g_u^{\text{pref}}, \mathcal{L}_u\}$                                 ▷ User Feedback
7:     $G = \text{Merge}(G_m, G_u)$
8:     $\mathcal{P} = \mathcal{F}(G \mid s^{\text{update}})$
9:     $\mathcal{L} = \mathcal{L}_m + \mathcal{L}_u; i = i + 1$
10: **end while**
11: **return** $\hat{I}, \mathcal{P} = \{\mathcal{P}^{\text{pip}}, \mathcal{P}^{\text{gly}}, \mathcal{P}^{\text{tex}}\};$

---

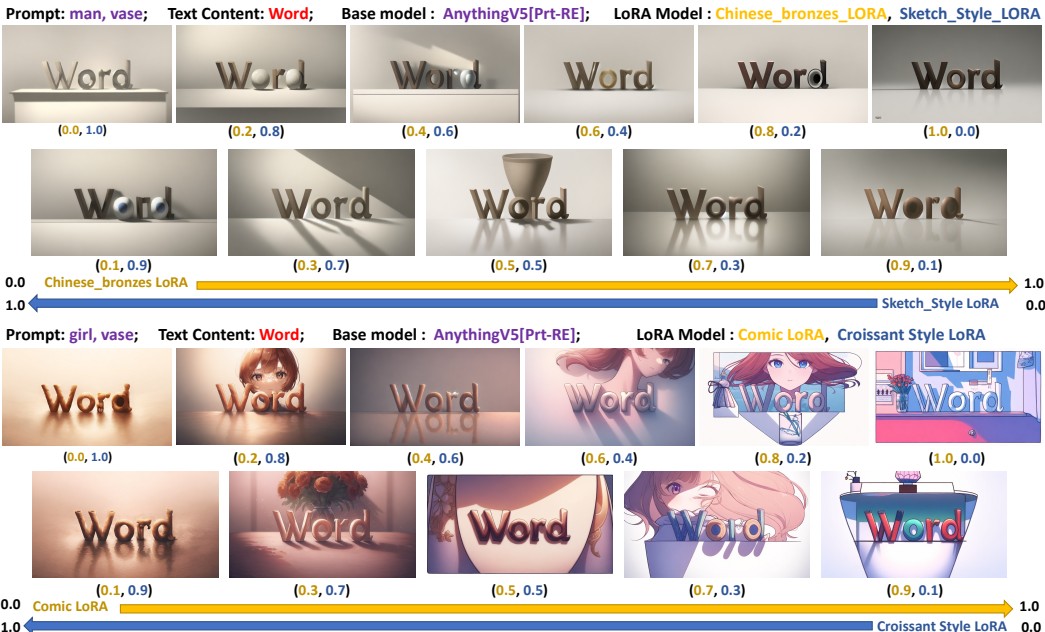

Figure 12: Effect of merging LoRAs with different weights. The top example fuses **Chinese Bronzes** and **Sketch Style** LoRAs; the bottom fuses **Comic** and **Croissant Style** LoRAs.

- **Thoughts-4th (Keywords: "Oil Painting", "Van Gogh", "Monet", "RealisticOilPainting")**

  - **Vocabulary:** Straightforward scene without explicit ties to any single artist or style.

  - **Theme:** A possibly everyday or spiritual moment suitable for an oil painting approach.

  - **Content:** Two figures in a church—a scene often depicted in a realist or classical oil painting tradition.

  - **Implicit Culture:** Western context, aligning well with artists like Van Gogh or Monet, although the prompt is not artist-specific.

  - **Reader Perception:** Could be viewed as an ordinary, contemplative moment in a church, commonly illustrated in oil paintings.

  - **Summarization:** "Oil Painting" is broad enough to encompass the scene without restricting it to a specific artist or ultra-realistic style.

  - **Selected:** *Oil Painting*

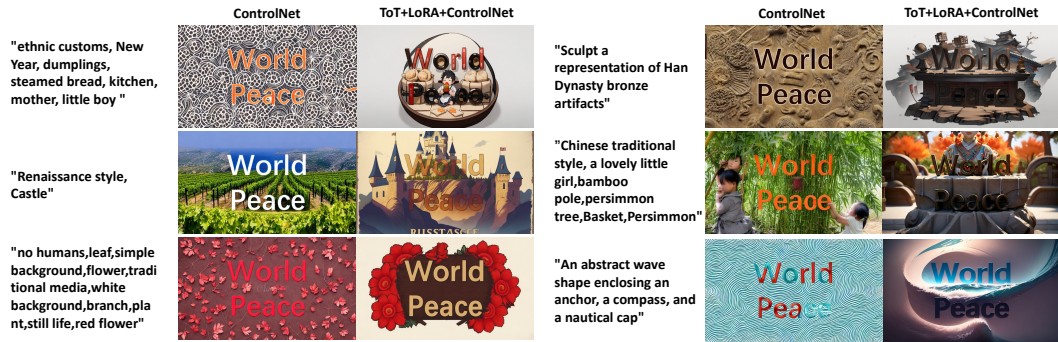

Figure 13: Comparison of WordArt texture rendering on the glyph "World Peace" (ControlNet vs. ToT-GPT4-LoRA+ControlNet).

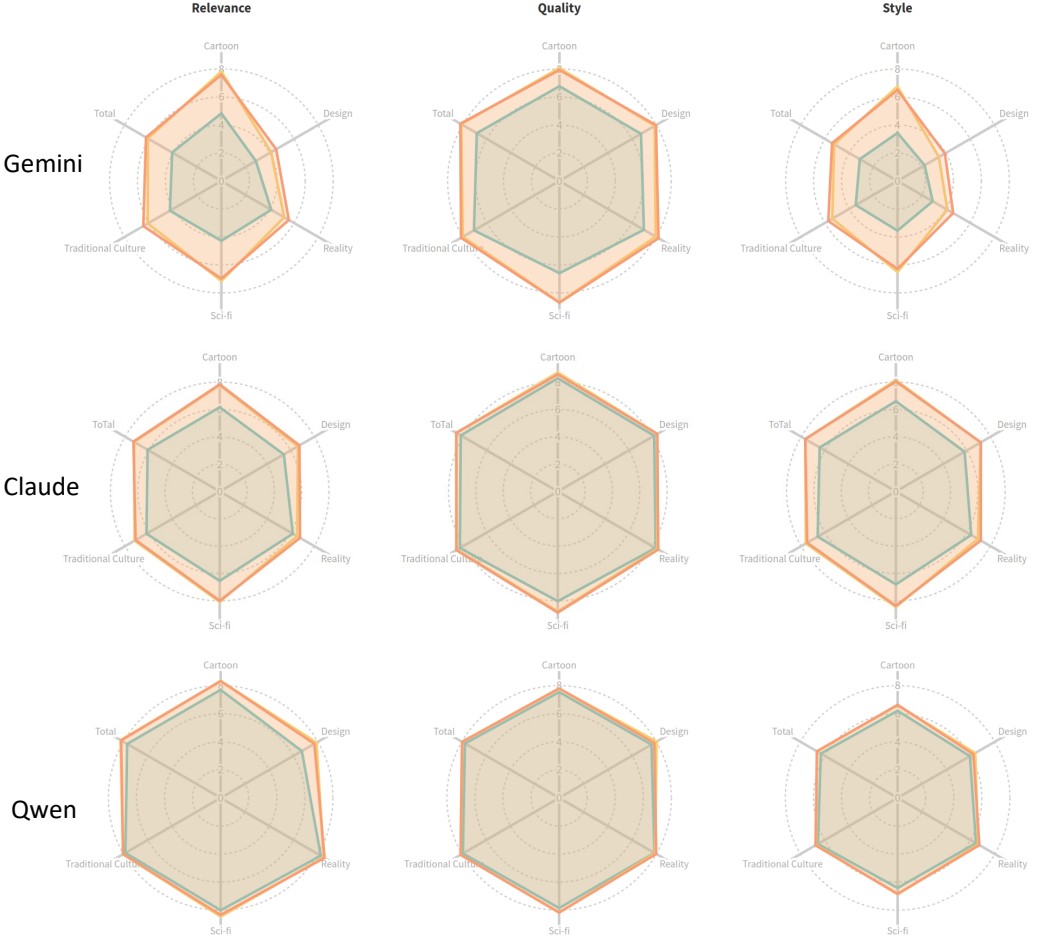

Figure 14: A radar-chart comparison of WordArt rendering on the glyph "World Peace" (ControlNet vs. ToT-GPT4-LoRA+ControlNet).

## C    MORE DETAILS OF IMAGE EVALUATION

### C.1    TOT-LORA EVALUATION PROMPT

*You are now a scoring judge for a generative model. I will show you images and prompts produced by the model. Please assess how well each image's content aligns with the corresponding prompt and rate its overall image quality on a scale of 1–10. Provide three numerical scores: (1) **Prompt***

*Relevance*, (2) *Image Quality*, and (3) *Style–Prompt Match*. *Return only these numbers without any further explanations.*

## C.2   LLAVA EVALUATION PROMPT

*Given an input sentence, break it down into real-world, visually identifiable "Targets." Use the template below to list these targets:*

```
Targets:{target, target, ...}
<Input sentence>:  {input}
```

*Subsequently, for each listed target, query LLaVA to check whether it appears in the image by asking: "Is {target} present in the photo? Please answer Yes or No."*

## D   MORE DETAILS OF WORDART DATASET

To thoroughly evaluate the *MetaDesigner* framework, we assembled 150 prompts spanning five themes—*cartoon*, *design*, *reality*, *sci-fi*, and *traditional culture*. These themes encompass a broad array of artistic styles and thematic elements, challenging *MetaDesigner* to produce aesthetically and contextually diverse WordArt.

**Linguistic Diversity and Selection Criteria.** From this larger set, we selected 20 prompts for a user study, specifically chosen to represent English, Chinese, Japanese, and Korean. This subset was curated to ensure linguistic variety, thematic breadth, and real-world applicability (e.g., e-commerce, education, and digital content creation).

**Statistics and Analysis.** Each of the five themes contains 30 prompts, ensuring balanced coverage of different styles and subject matter. We considered cultural relevance, popularity in digital art, and commercial or educational applicability when selecting these prompts. Figure 9 highlights diverse use cases such as cartoon text, Chinese surnames, and solar terms, illustrating the dataset's capacity to accommodate varied artistic and linguistic expressions.

**Applicability and Use Cases.** The dataset demonstrates utility for numerous domains. In e-commerce, for instance, customized WordArt can enhance user engagement; in educational contexts, multilingual text designs can enrich language learning and cultural instruction. By testing *MetaDesigner* in these settings, we validate both its ability to generate contextually and culturally relevant WordArt and the dataset's suitability for studying the intersection of AI, language, and art.

**WordArt Dataset Analysis.** Figure 15 illustrates numerical data from our "Image Plaza," where over **two million** images have been generated. These images feature a range of aspect ratios, word counts, and thematic content, and are retrievable in our WordArt space[2].

Figure 16 depicts user-preference analytics, revealing that over one million images have been produced by users. The majority favor a 16:9 aspect ratio, typically using up to five characters of text. Moreover, alpha-channel images are especially popular.

Table 3: **VLM evaluation** in two dimensions: *Aesthetics* and *Creativity*, scored from 0–10 (higher is better). Ours[*] indicates a version without agents.

| Evaluation | SDXL | SDXL-Aug | TDiff | TDiff-2 | Anytext | DALLE 3 | Ours[*] | **Ours** |
|---|---|---|---|---|---|---|---|---|
| Aesthetics | 7.9 | 8.1 | 6.1 | 6.4 | 7.0 | 8.5 | 8.3 | **8.7** |
| Creativity | 7.5 | 7.7 | 5.8 | 6.0 | 6.1 | 7.6 | 7.7 | **7.9** |

---

[2]`https://modelscope.cn/studios/WordArt/WordArt`

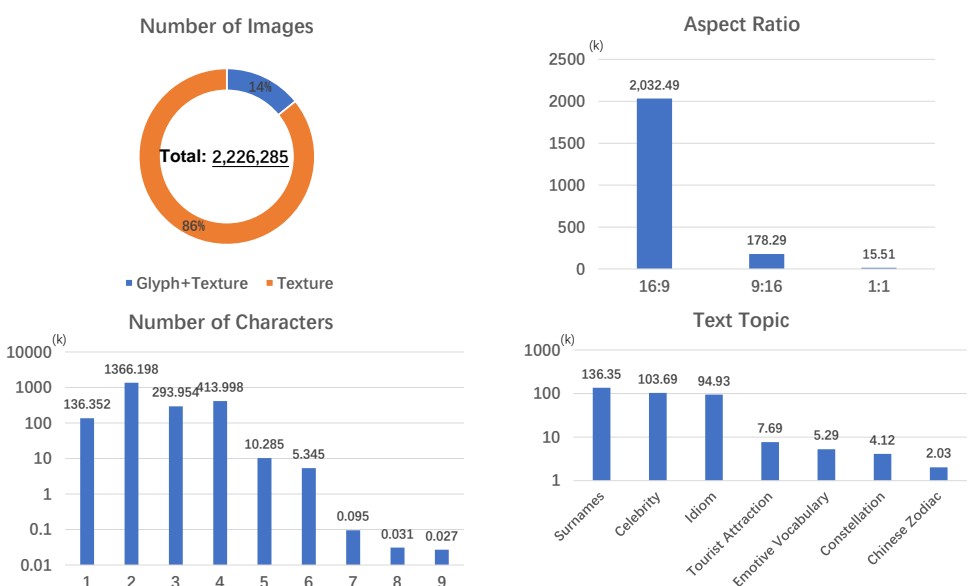

Figure 15: Numerical analyses of our WordArt dataset. The horizontal axis denotes the number of images (in thousands), while the vertical axis represents category labels.

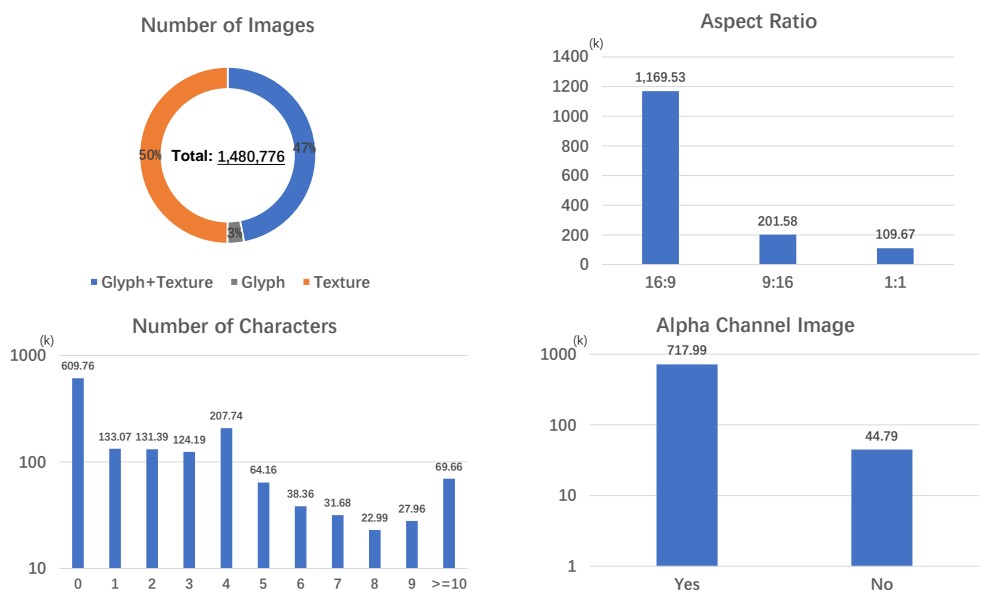

Figure 16: User preference analytics. The horizontal axis shows the number of images (in thousands), and the vertical axis displays specific categories. "0" in the "Number of Characters" panel indicates use of a mask image rather than textual input.

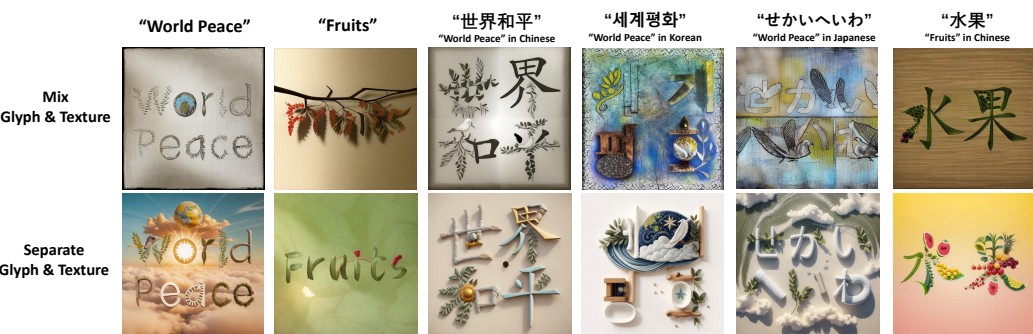

Figure 17: Additional examples comparing different methods.

Figure 18: Visual comparison of merging vs. separating the Glyph and Texture Designers.

**LLaVA response**

**GPT-4V response**

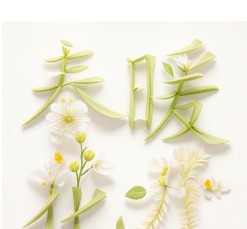

**Consistency** : "score": 9; "reason": "The image contains all the required elements, such as flowers, young leaves, sunlight, warmth, curves, petal patterns, grass texture, ripples, warm tones, and vitality. The elements are properly positioned and contribute to the overall composition."

**Style** : "score": 9; "reason": "The image has a warm and vibrant color tone, which matches the requested artistic style. The sunlight and warmth elements add to the overall atmosphere, creating a sense of vitality and life. The use of curves and petal patterns adds visual interest and contributes to the artistic innovation."

**Quality**: "score": 9;"reason": "The image is clear and well-composed, with good lighting effects that enhance the overall visual appeal. The technical excellence is evident in the details of the flowers, leaves, and grass texture, which are all well-executed."

**Creativity** : "score": 8; "reason": "The image showcases a high level of artistic innovation through the use of various elements, such as the sunlight, warmth, and grass texture. The combination of these elements creates a unique and visually appealing composition that stands out from more conventional floral images."

**Consistency** : "score": 8; "reason": "The image effectively incorporates several elements from the prompt, including flowers and leaves, and showcases a sense of warmth and vitality. However, it lacks direct representation of sunlight and a textural representation of grass."

**Style** : "score": 9; "reason": "The overall aesthetic aligns well with the prompt. It features warm tones and curves in the arrangement of flora, creating an inviting atmosphere. The minimalist background enhances the focus on the floral elements, contributing to the intended style."

**Quality**: "score": 9; "reason": "The image exhibits high clarity with a well-considered composition. The lighting effectively highlights the textures and colors of the flowers, adding to the technical excellence of the photograph."

**Creativity**: "score": 8; "reason": "The arrangement of flowers to form characters is a unique artistic interpretation, showing ingenuity in using natural elements to convey a message. The innovation in presentation is commendable, though it remains within established creative boundaries."

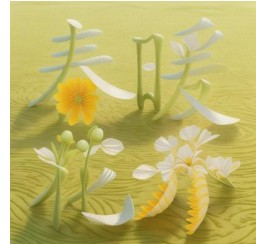

**Consistency**: score": 9; "reason": "The image contains all the required elements, such as flowers, young leaves, sunlight, warmth, curves, petal patterns, grass texture, ripples, warm tones, and vitality. The elements are well-positioned and contribute to the overall composition of the image."

**Style**: "score": 9; "reason": "The image matches the requested artistic style, color tone, and atmosphere. The warm tones, sunlight, and warmth create a sense of vitality and life, while the curves and petal patterns add an artistic touch. The grass texture and ripples contribute to the overall visual interest of the image."

**Quality**: "score": 9; "reason": "The image is of high quality, with clear details and a well-composed composition. The lighting effects are well-executed, and the technical aspects of the image are excellent."},

**Creativity**: "score": 8; "reason": "The image is quite creative, with a unique combination of elements that create a visually interesting and engaging composition. The use of warm tones, sunlight, and warmth adds a sense of vitality and life to the image, while the curves and petal patterns contribute to its artistic appeal."

**Consistency**": "score": 9; "reason": "The image features a variety of flowers and young leaves arranged in a way that is visually coherent and interpretable. All elements from the prompt, such as flowers, leaves, and natural textures, are present and well-integrated, creating a harmonious composition."

**Style** : "score": 8; "reason": "The color tone is warm, with yellows and greens dominating, which matches the request for warm tones. The overall atmosphere feels vibrant and lively, aligning well with the concept of warmth and vitality, although the artistic style may not fully resonate with a traditional floral representation."

**Quality**: "score": 9; "reason": "The image is clear, with sharp details that highlight the textures of the flowers and leaves. The lighting is soft and enhances the overall warmth of the composition, contributing to a visually appealing and technically excellent photo."

**Creativity**: "score": 9; "reason": "The arrangement of the floral elements in a creative and artistic way demonstrates a high level of innovation. The use of petals and leaves to form an artistic representation shows uniqueness and a strong creative vision, setting it apart from typical floral imagery."

Figure 19: Differences between LLaVA- and GPT-4V-based evaluations.

**User Prompt :** **I want a peace of WordArt with text "西湖龙井"**   *Glyph Designer*

**Pipeline Designer**

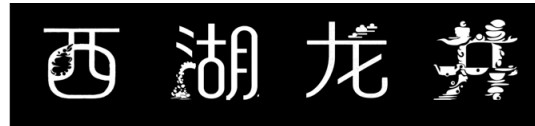

**Glyph Prompt**: teapot, tea, ripple, willow, water drop.

**Texture Prompt** :spring , green tea, mist, ripples, young leaves teapot, tea, ripple, willow, water drop

**Texture Designer**

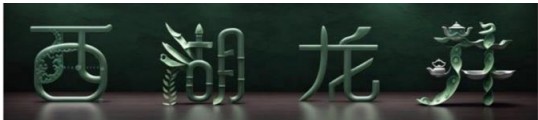

**ToT LoRA Selection**

**Vocabulary**: The prompt indicates a possible combination of realistic textures for portraying elements like the blooming green tea, the slate road, the mist, ripples, young leaves, teapot, tea, willow, and water drop. There is also hints of some abstract concepts that could be explored through artistic technique such as the texture effects on the slate road or the ripples of the water.

**Theme**: In terms of theme, the prompt portrays a calm, peaceful atmosphere typically associated with traditional tea ceremony scenes. This suggests a leaning towards symbolism related to tranquility, tradition, and nature.

**Content**: The colors that can be inferred from the prompt suggest a fresh, outdoorsy, and calm mood. The green and misty attributes lean towards a specific color palette that can be found within nature-oriented artwork.

**Implicit Culture**: The elements mentioned in the prompt are associated with Asian or Eastern cultures, especially due to the emphasis on tea ceremony symbolism.

**Reader Perception**: Given that the scene being painted invites a thorough exploration of different textures like the softness of the tea leaves, the hardness strength of the slate road, the delicate nature of the mist, etc., it clearly emphasizes the use of realistic rendering techniques.

**Summarization**: Considering all the above analysis, the prominent elements from the prompt appear to be 'traditional culture' and 'realistic'. The theme of the scene, the implied cultural context, and the necessity for realistic texture rendering contribute to these choices.

**Selected : realistic**

**Vocabulary**: The prompt contains vocabulary indicative of a realistic scene. Words like 'spring', 'green tea', 'slate road', 'mist', 'ripples', 'young leaves', and 'water drop' contributes to setting a particular atmosphere that portrays a concrete moment in time, much akin to a realistic scene.

**Theme**: The theme here is a serene springtime moment experienced while having tea. Green tea, willows, and young leaves symbolize fresh beginnings, calm and tranquility presented in a realistic scene.

**Content**: The color palette inferred from the prompt portrays a natural, refreshing, soothing mood; spring greens, slate grays, mist white or soft blues, and the warmth of tea. The Mood is peaceful and calm, often associated with realistic scenes.

**Implicit Culture**: The cultural context is drawn from tea-drinking cultures, where spring green tea reflects East Asian cultural traditions like Chinese or Japanese, which prefer doing it in natural sceneries, aligning it more towards a realistic scene.

**Reader Perception**: The artistic techniques implied through the objects e.g., blooming tea, road with mist, young leaves, water drops, are more nature-centered, together creating a coherent, relatable, and realistic scene.

**Summarization**: In summation, the described scenario in the input prompt appears to be painting a picture of a particular moment in nature - note the spring blooms, the mist, the slate road, the tea, hinting towards a realistic scene rather than focusing on a single realistic object.

**Selected : realisticscene**

**Evaluation**

**Consistency** : **"score": 8, "reason": "**The image contains a teapot, tea, ripple, willow, water drop, and young leaves, which are all elements of the prompt. The arrangement of these elements is visually appealing and well-organized.**"**

**Style** : **"score": 9, "reason": "**The image has a green tea theme, which is consistent with the prompt. The color tone is predominantly green, and the atmosphere is calm and serene, which aligns with the requested artistic style**."**

**Quality**: **"score": 9, "reason": "**The image is of high quality, with clear details and a well-composed arrangement of the elements. The lighting effects are subtle and effective, enhancing the overall visual appeal of the image.**"**

**Creativity** : **"score": 8, "reason": "**The image is creative in its presentation of the elements, with the teapot and tea leaves arranged in a visually interesting manner. The use of ripples, willow, and water drop adds a touch of artistic flair to the composition**."**

Figure 20: Overview of the full *MetaDesigner* pipeline.

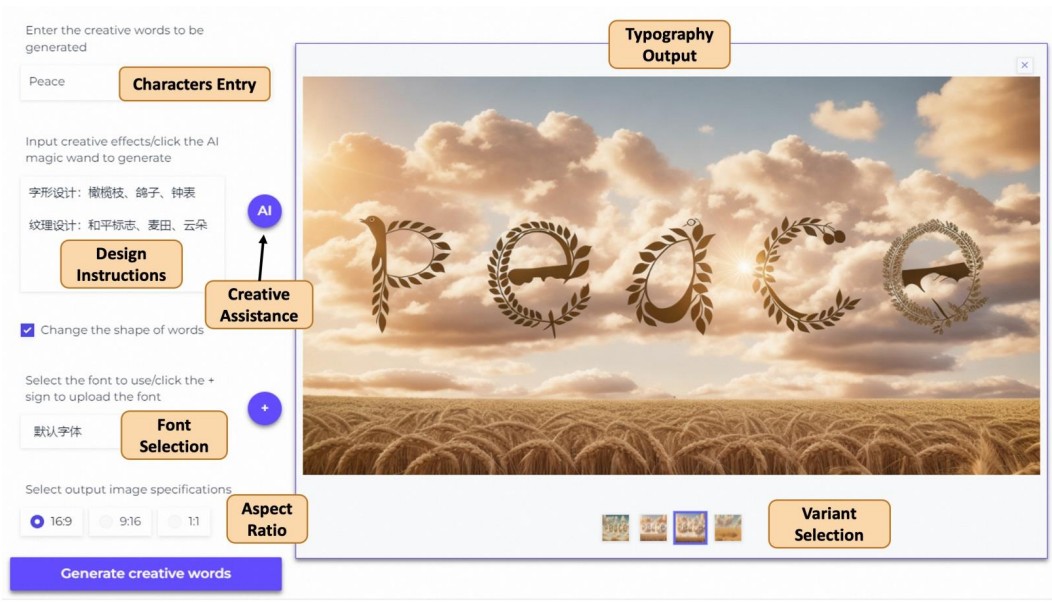

Figure 21: User interface of the *MetaDesigner* system.

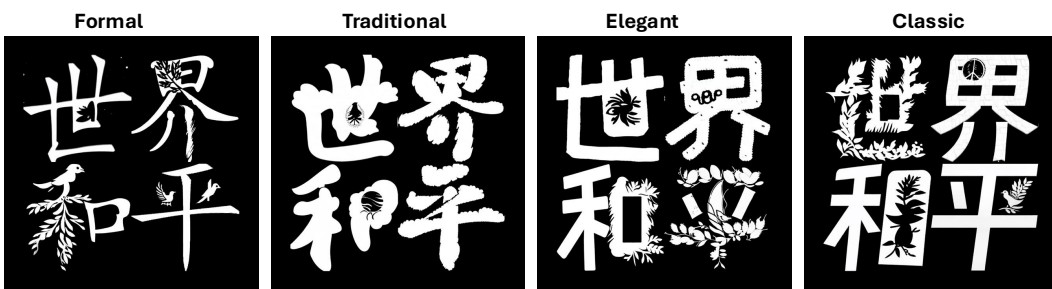

Figure 22: Impact of font type on glyph transformations.

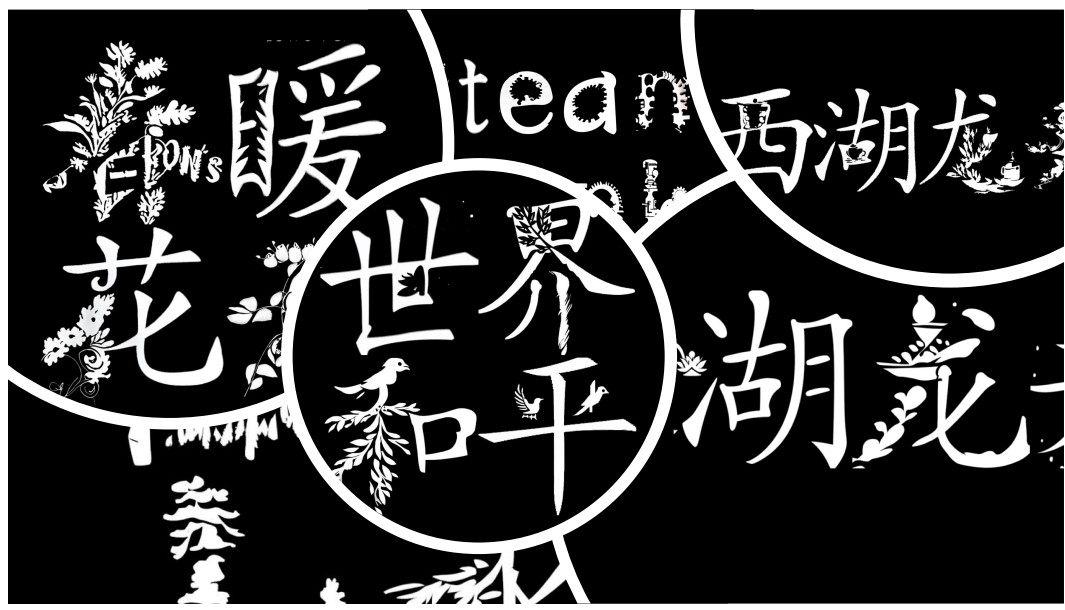

Figure 23: Further examples generated by the *Glyph Designer*.

Table 4: Comparison of ToT-Selection using different models.

| Evaluation Theme | GPT-4 | GPT-4V |
|---|---|---|
| World Peace | design -> poster (x3)
traditional culture -> oil painting (x2) | traditional culture -> oil painting (x1)
general -> general (x2)
realistic -> realisticscene (x1)
traditional culture -> papercut (x1) |
| Colorful World | sci-fi -> future world (x2)
traditional culture -> oil painting (x1)
traditional culture -> inkpainting (x2) | sci-fi -> future world (x1)
traditional culture -> oil painting (x1)
traditional culture -> papercut (x1)
realistic -> realisticscene (x1)
sci-fi -> cyber (x1) |
| Artistic Style Fusion | traditional culture -> oil painting (x3)
traditional culture -> engrave (x1)
design -> poster (x1) | design -> poster (x2)
traditional culture -> inkpainting (x1)
traditional culture -> chinesecomic (x1) |
| Future Technology | sci-fi -> cyber (x5) | sci-fi -> cyber (x5) |
| Children's Fun | cartoon -> 3dcartoon (x4)
design -> poster (x1) | cartoon -> 3dcartoon (x4)
cartoon -> forchildren (x1) |

# E  EFFECT OF GPT-4/GPT-4V ON ToT-SELECTION

We evaluated the GPT-4 and GPT-4V models on 25 test prompts, divided into five thematic categories. Table 4 shows the ToT-Selection results. Compared to the more constrained *Future Technology* theme, *World Peace* and *Colorful World* allow broader interpretations. On these open-ended themes, GPT-4V produces more varied outcomes, indicating a nuanced grasp of user intent. However, for the more focused *Future Technology* theme, GPT-4V tends to be more consistent than GPT-4. Details of the test prompts within each theme are outlined below:

- **World Peace Theme**

    - *"watercolor dove, olive branch, rainbow blend, peace symbol"*
    - *"multicultural children, circle dance, traditional patterns, vibrant clothing"*
    - *"hands joining together, global unity, soft pastels, harmony symbols"*
    - *"peace garden, blooming flowers, meditation space, tranquil atmosphere"*
    - *"world flags blending, peaceful doves flying, sunrise hope, gentle clouds"*

- **Colorful World Theme**

    - *"earth view, kaleidoscope, abstract continents, space art"*
    - *"butterfly swarm, gradient sky, cultural patterns, colorful wings"*
    - *"rainbow coral reef, tropical fish, underwater prism, vibrant marine life"*
    - *"northern lights aurora, starry night, color spectrum, celestial dance"*
    - *"spring festival lanterns, color explosion, festive celebration, dynamic lights"*

- **Artistic Style Fusion Theme**

    - *"art nouveau, digital wireframe, modern fusion, geometric patterns"*
    - *"ukiyo-e waves, pixel art, japanese digital, style blend"*
    - *"renaissance painting, neon aesthetics, classical meets cyberpunk"*
    - *"baroque architecture, minimalist overlay, historical modern mix"*
    - *"tribal patterns, glitch art fusion, ancient digital blend"*

- **Future Technology Theme**

    - *"cybernetic cityscape, holographic interface, neon circuits, quantum particles"*
    - *"robotic augmentation, bioluminescent tech, neural networks visualization"*
    - *"quantum computing visualization, data crystals, energy flows"*
    - *"cyborg nature fusion, techno-organic growth, synthetic biology"*
    - *"artificial intelligence mindscape, digital consciousness, virtual reality portals"*

- **Children's Fun Theme**

    - *"playful teddy bears, rainbow playground, bubble floating, magical toybox"*

- *"cartoon dinosaurs, candy colored clouds, children's storybook, whimsical adventure"*
- *"magical treehouse, flying paper planes, fairy lights, childhood dreams"*
- *"circus animals parade, balloon animals, cotton candy skies, cheerful carousel"*
- *"fantasy schoolyard, animated crayons, floating books, imagination sparkles"*

