# OpenReview forum: "MetaDesigner: Advancing Artistic Typography through AI-Driven, User-Centric, and Multilingual WordArt Synthesis"
_ICLR.cc/2025/Conference — ICLR 2025 Poster_

### Official Review · Reviewer_9F4p · 2024-11-01

**Soundness:** 2
**Presentation:** 2
**Contribution:** 2
**Rating:** 6
**Confidence:** 4

**Summary:**

The paper studies the problem of generating stylistic text (with stylized glyph and texture) from input text prompts.

The core contribution is a multi-agent system that factors text generation into multiple sub-tasks, and addresses them with specialized LLM, LMM and stable diffusion agents, including: 1)  a pipeline designer for generating design programs to instruct other agents; 2) a glyph designer for glyph generation; 3) a texture designer for texture generation;  4) a Q&A evaluator that gathers user and LMM feedback for hyperparameter tuning of other agents to iteratively improve output quality.

**Strengths:**

1. Artistic text generation is an important problem, whose solutions can be useful in many practical scenarios.

2. The amount of the contribution is non-trivial since a lot of effort has gone into adapting and connecting different large models together to form a complete, large working system.

3. The whole system is well motivated and designed.

**Weaknesses:**

1. Some technical details are missing. For example, in “Normal & Traditional Glyph”, the rendering details of normal and traditional glyphs based on $s^{gly}$ are unclear. Also, it is not clear how vector graphics optimization in “Semantic Glyph Transformation” is done. Where do the “target objects” come from and what is the loss function exactly? In Section 3.4, what is the meaning of $L_m$ and $L_u$ ?

2. Some experimental settings are questionable. First, comparison to SDXL is unfair. Traditional text-to-image models like SDXL often require detailed prompts and may not work well with brief instructions (e.g., “Create a stylish word World Peace representing its meaning”) that the proposed system is supposed to handle. The proposed system utilizes GPT for input prompt extension and, thus, it should be compared against a GPT-augmented SDXL where GPT is used to extend the input prompt into a more detailed description relevant for text generation.  Second, in the user study, “Aesthetics & Creativity” should be divided into two independent dimensions, instead of being merged as a single one. Third, the claim “Table 2 presents quantitative results using SSIM and LPIPS metrics, further reaffirming the excellence of our approach in terms of readability and aesthetics.” is inadequate. SSIM and LPIPS mainly measure image similarity and, therefore, are not suitable for evaluating readability. To compute quantitative scores for readability and legibility, a more reasonable way may be to apply a model to detect and recognize text on the generated images and then compare the outputs with the ground truth text. Last, the “quantitative” evaluation of the Q&A evaluation agent is lacking. The results of the proposed method, without this agent, should be added to Table 1 and Table 2.

3. Some results are not satisfactory. As shown in Figure 5, the outputs by the proposed method, although being visually appealing, suffer from low text legibility and readability: some letters and words are either hard to recognize due to some of their strokes being excessively deformed or stylized (e.g., as leaves), or hard to read because of low color contrast between them and the background.

**Questions:**

1. Should Equation (13) use max instead of argmax?

2. For the hyperparameter optimization in Section 3.4, it seems that the parameters are updated with GPT (as illustrated in Figure 4). What are the detailed prompts used to instruct GPT to output the updated prompts/parameters for each case in Figure 4?

3. For the user study, what does “Text accuracy” mean? Was the input text exposed to the participants during the study? What is the meaning of the percentages in Table 1?

---

> ### Author Response · Authors · 2024-11-25
> **Response to the Weakness**
>
> ### Response to the Weakness:
> 1) Thank you for your valuable feedback. $s^{gly}$ represents the extended prompt by the Pipeline Agent,
>
> - Regarding normal and traditional glyphs, we categorized glyphs based on their stylistic attributes. "Normal glyphs" refer to modern fonts commonly used in printed texts, such as Song and Hei typefaces. In contrast, "traditional glyphs" encompass Chinese calligraphy styles like Li, Xing, and Cao scripts, characterized by their historical and artistic significance. To illustrate the distinction, we will provide visual examples in the manuscript.
>
> - For Semantic Glyph Transformation, the Glyph Agent employs differential rendering techniques to modify target glyphs [1,2]. The Glyph Designer deforms the glyph based on the semantic concept provided by the pipeline designer. We utilize both score diffusion loss and CLIP loss to guide the deformation process, ensuring that the glyph evolves toward the desired semantic concept. This optimization process typically requires 80 -180 iterations (can be set by the user/agent), depending on the desired level of deformation. We set to differential rendering size to 350x350 to reduce the time cost. The loss function is a score diffusion loss, which ensures that the generated glyphs maintain content consistency while aligning with the desired semantic concept.
>
> In Section 3.4,  $\mathcal{L}_{u}$ and $\mathcal{L}_{m}$ represent the objective loss (average score) from the user and LLaVA model feedback respectively.
>
> 2)  Thanks for your valuable feedback, we follow your suggestion to conduct further studies:
>    - GPT-Augmented SDXL: Actually, for all methods in Figure 5, we employed the same prompt. Following your suggestion, we have added GPT-Augmented SDXL to the manuscript and correspondingly included the visualization results in the SOTA comparison, denoted as `SDXL-Augmented/Aug.'. Our experiments revealed that while the more detailed prompts provided by GPT augmentation can enhance image quality, they do not significantly improve the correctness of SDXL's text generation. We surmise that this is primarily due to the inherent limitations in SDXL's comprehension of text.
>
>    - Divide "Aesthetics & Creativity" into two dimensions: We have divided "Aesthetics & Creativity" into two distinct dimensions and conducted a new user study. The results are presented in the manuscripts.
>
>    - Regarding the discussions about SSIM and LPIPS in Table 2,  While the SSIM and LPIPS metrics can partially reflect the spatial structural similarity between the generated images and the ground truth images, they do not fully account for the readability of the text. We have also attempted to employ OCR to assess readability. However, we found even the most advanced OCR methods, such as PP-OCRv4, to have a low recognition rate for the artistic fonts generated by our approach as well as other methods, rendering it an unreliable means of evaluation. Consequently, we have introduced the evaluation dimension of "Text Accuracy" in our user study to manually assess the readability of the text in the generated images. We will elucidate the meaning of `Text Accuracy' in our subsequent responses.
>
>  - Results of the proposed method, without a relative agent:  We have included "ours (w/o agents)" in both the user study and numerical experiments. For details, please refer to our manuscript. Additionally, we have included the corresponding visualizations. We have also attempted to employ OCR to assess readability. However, we found even the most advanced OCR methods, such as PP-OCRv4, to have a low recognition rate for the artistic fonts generated by our approach as well as other methods, rendering it an unreliable means of evaluation. Consequently, we have introduced the evaluation dimension of "Text Accuracy" in our user study to manually assess the readability of the text in the generated images. We will elucidate the meaning of "Text Accuracy" in our subsequent responses.
>
> 3)  Thanks for your valuable feedback. The creativity and the readability of the text indeed require a balance. If users prioritize ensuring readability, this can be achieved through the following two methods: 1) In Glyph Designer, by reducing the deformed stroke areas to retain more of the original stroke information; 2) In Texture Designer, by increasing the control scale of the ControlNet to enhance the clarity of the text edges, thereby improving readability. This also incorporates user feedback to align with user preferences.
>
> [1] Word-As-Image for Semantic Typography. ACM Trans. Graph. 42(4): 151:1-151:11 (2023)
>
> [2] WordArt Designer: User-Driven Artistic Typography Synthesis using Large Language Models. EMNLP 2023: 223-232

---

> > ### Comment · Reviewer_9F4p · 2024-11-26
> >
> > The rebuttal has addressed most of my concerns, but my concern on the problems in the visual results remains. I finally decide to raise my score from 5 to 6.

---

> ### Author Response · Authors · 2024-11-25
> **Response to the Question**
>
> ### Response to the Questions:
> 1) We aim to achieve a dual objective: synthesizing an image with the highest evaluation score while simultaneously identifying the optimal hyperparameters$P^{gly}$ and $P^{tex}$. To this end, we employ the argmax function to optimize the hyperparameter settings that maximize the evaluation score.
>
> 2)  Optimizing Prompts .
> ```
> f"""As a prompt optimization expert, please improve the
>     following prompt based on the evaluation results.
>         Original Prompt:
>         "{original_prompt}"
>
>         Evaluation Results:
>         Consistency Score: {evaluation['consistency']['score']}
>         Reason: {evaluation['consistency']['reason']}
>
>         Please optimize the prompt following these rules:
>         - Duplicate important nouns
>         - Add emphasis using () brackets for key elements
>         - Add specific spatial relationships
> """
>
> f"""As a prompt optimization expert, please improve the
>     following prompt based on the evaluation results.
>         Original Prompt:
>         "{original_prompt}"
>
>         Evaluation Results:
>         Style Score: {evaluation['style']['score']}
>         Reason: {evaluation['style']['reason']}
>
>         Please optimize the prompt following these rules:
>         For low style scores, extract the key style elements from prompt & reseaon
>         - Style: Anime, Cartoon, Realistic, Cyberpunk, Cybernetic, Cybernetic, Cybernetic
>         - Enhance style-related descriptors, .
>         - Add specific artistic technique references
>         - Strengthen atmosphere descriptions
> """
>
> f"""As a prompt optimization expert, please improve the
>     following prompt based on the evaluation results.
>         Original Prompt:
>         "{original_prompt}"
>
>         Evaluation Results:
>         Quality Score: {evaluation['quality']['score']}
>         Reason: {evaluation['quality']['reason']}
>
>         Please optimize the prompt following these rules:
>         For low quality scores, add technical specifications/keywords following the prompt & reason
>         - Resolution: 8K, 4K, High resolution
>         - Details: Highly detailed, Intricate, Minute details
>         - Lighting: 4K lightning, Natural light, Artificial light, Backlit, Spotlight, Ambient light
> """
>
> f"""As a prompt optimization expert, please improve the
>     following prompt based on the evaluation results.
>         Original Prompt:
>         "{original_prompt}"
>
>         Evaluation Results:
>         Creativity Score: {evaluation['creativity']['score']}
>         Reason: {evaluation['creativity']['reason']}
>
>         Please optimize the prompt following these rules:
>         For low creativity scores:
>
>         - Reset the parameters from the original prompt and reason:
>             - Glyph Parameters
>                 - FontTyle as follow ,semantic is more creativity:
>                     Formal
>                         - Formal,
>                         - Elegant,
>                         - Classic,
>                         - Traditional
>                     Semantic
>
>                 - Iterations: 80~180, the higher the more artistic
>             - update the texture prompt
>                 - Add unique viewpoints
>                 - Include unexpected elements
>                 - Enhance artistic innovation aspects
> """
>
> ```
> All the feedbacks are strings that will be further parsing by the GlyphDesigner and TextureDesigner.
>
> 3) "Text accuracy" is used to reflect whether the text in the images is written correctly. Specifically, during the user study process, we present users with six images (corresponding to the six methods in Table 1) and the input text, then prompt them with the question: "Select the images where you believe the text is written correctly (multiple choices allowed)." The percentages in Table 1 represent the proportion of times each method was chosen by the users. It is worth noting that the "Text accuracy" question allows multiple selections, hence the sum of all values in the second row of Table 1 does not equal 100. In contrast, the "Aesthetics & Creativity" question is a single-choice question, so the sum of all values in the third row of Table 1 exactly equals 100.

---

### Official Review · Reviewer_zqVY · 2024-11-02

**Soundness:** 2
**Presentation:** 3
**Contribution:** 2
**Rating:** 6
**Confidence:** 3

**Summary:**

This paper introduces a multi-agent system capable of generating WordArt based on user instructions, with flexibility in style and texture. The system primarily consists of a Pipeline Designer, a Glyph Designer, and a Texture Designer, each responsible for overall process design, generating font images, and applying background and text textures, respectively. A feedback mechanism is integrated into the system, where prompt refinement is achieved through interactions between GPT and LLava or users, improving prompt to produce higher-quality images that better align with user expectations.

**Strengths:**

1. The WordArt examples generated by the model in this paper demonstrate higher visual quality, more accurate text representation, and greater consistency in overall style compared to previous methods.

2. The system proposed in this paper can generate WordArt in diverse styles, closely resembling designs created by human designers, owing to the use of a style LoRA library. Additionally, it demonstrates the effect of merging LoRAs with varying weights.

3. The paper utilizes advanced prompt-LM techniques, such as Chain-of-Thought, Tree-of-Thought, and Feedback Refinement, to enhance the accuracy of sub-prompts for each agent, resulting in generated content that better aligns with user preferences.

**Weaknesses:**

1. **Lack of novelty**: The improvements in “Accuracy and Creativity” claimed in the paper primarily depend on the construction of the style LoRAs library in “Texture Designer”, which aligns with previous experience. The impact and novelty of other modules are also insufficiently highlighted. The paper adopts a style closer to a technical report.

2. **Insufficient Experimental Design**: For example, (a) the user study includes only 11 participants, and it may benefit from a fair comparison using publicly available VLMs (e.g., GPT4-V); (b) the ablation study only analyzes the ToT module within the “Texture Designer,” without examining the analyses of other modules; (c) the comparison of generated results with other models is limited to 6 samples in Fig. 5, and more samples would be expected.

**Questions:**

I'd like to pose questions that can address the weaknesses discussed.
1.  **More Detailed Examples of the Full Pipeline**: While Fig. 10 presents the final results, it would be beneficial to show the output at each step of the system, such as the glyph images generated by the “Glyph Designer Agent.” Subfigures (like the picture with dark green tones in the lower left corner) may be good choices for illustrating these intermediate steps.

2.  **Details on the Construction of the Style LoRAs Library in the “Texture Designer”**: Additional information is expected regarding the number of trainable parameters, the volume of collected data, and a comparison of data size with previous WordArt synthesis works, among other relevant details.

3. **Information about the inference efficiency**: A comparison of image generation time and computational resources required, relative to other models

4. **Impact of Replacing with GPT4-V**: a) In both the “Texture Designer” and “Parameter Optimization” stages, image input is utilized. Would replacing the previous GPT-4 with GPT4-V be a more effective choice? Quantitative and sample-based analyses? b) What differences would arise if using GPT4-V instead of LLava in the “Feedback” stage?

5. **Potential Common Problems in Generated Images**: a) Although the images in Fig. 5, Fig. 8, and Fig. 9 may possess artistic qualities, the text color often resembles the background, making it less distinguishable compared to results from other models.  b) In the lower-left of Fig. 8, style information, such as “Renaissance,” is generated as part of the text—what causes this, and is this phenomenon common?

---

> ### Author Response · Authors · 2024-11-25
> **Response to the Weakness**
>
> ###  Response to the Weakness:
> 1) Thank you for your valuable feedback. While the construction of the style LoRAs library is indeed a significant component of our approach, it’s the novel combination of this technique with other innovative modules that drives the substantial improvements in "Accuracy and Creativity." Our primary goal is to empower non-professional users to create high-quality, creative WordArt with minimal effort. To achieve this, we introduce a multi-agent system that leverages the strengths of various techniques. Our proposed framework introduces several novel aspects:
>  - Enhanced Prompt Engineering: The Pipeline Agent leverages a large language model to extend user prompts into detailed descriptions, enabling more precise control over the generated WordArt.
>
> - Semantic-Aware Glyph Synthesis: The Glyph Agent introduces semantic deformable capabilities, allowing for flexible glyph transformations and improving the overall aesthetic appeal of the generated WordArt.
>
> -  Adaptive Texture Rendering: The Texture Agent employs a ToT (Tree-of-Though) selection scheme to dynamically choose the most suitable LoRA, ensuring alignment with user preferences and generating diverse and high-quality textures.
>
>  - Iterative Refinement with Feedback: The feedback mechanism allows for interactive refinement of the generated WordArt, enabling users to provide feedback and guide the synthesis process toward their desired outcome.
>
>
> By combining these techniques, our approach offers a comprehensive solution for generating creative and high-quality WordArt, even for non-professional users. We believe that our work pushes the boundaries of WordArt synthesis and provides a valuable contribution to the field. We will update the introduction and the details to highlight our contribution. We will update the manuscript to highlight the novelty.
>
> 2) Thank you for your valuable feedback. We have incorporated additional evaluation feedback studies and examples into the current uploaded version of the manuscript. The effect of the evaluation is updated in Table 1, and in Table 2; the VLMs evaluation results are listed in Table 3. More exemplary case is listed in Figure 17

---

> ### Author Response · Authors · 2024-11-26
> **Response to the Questions**
>
> ###  Response to the Questions:
> 1) We have incorporated additional intermediate results and analyses in the full pipeline into the final version of the manuscript, which will be uploaded shortly.
>
> 2) "Texture Designer" does not involve training any additional parameters. Instead, we curated a library of 68 pre-trained LoRA weights, which are loaded at inference time to adapt the model to specific user requests. To efficiently select the optimal LoRA, we implemented a Tree-of-Thought strategy. This approach leverages user-provided prompts (and their extended versions generated by the Pipeline Agent) to identify the most relevant LoRA weights. The entire LoRA library, originally 34GB in size, can be compressed to less than 17GB using fp16, enabling seamless loading into memory. The negligible switching time (~ 0.15s) between pre-loaded LoRAs enables efficient processing of four images within 8-9 seconds.
>
> 3) Information about the inference efficiency}}: A comparison of image generation time and computational resources required, relative to other models:
>
>   | Method                   | Iteration                  |   Render | Total |
>   | :-------------------: | :-------------------: |  :---------: | :-------: |
>   | Word As Image [1]      | 500 s                      | -              | 500s |
>   | DS-Fusion [2]              | 300$\sim$500 s      | -             | 300 $\sim$ 500 s |
>   | WordArt Designer [3]   | 36 s                        | 10s           | 46 s |
>   | MetaDesigner (Ours)         | 26 s                       | 10s           | 36 s |
>
> The MetaDesigner excels in inference efficiency, especially for single-round scenarios. The PipelineDesigner and evaluator complete their tasks in a mere 1$\sim$3 seconds, while the GlyphDesigner and TextureDesigner require approximately 26 and 10 seconds, respectively. Even when considering 3-5 rounds of iteration for high-quality WordArt synthesis, the total time cost is still under 2-4 minutes, surpassing the efficiency of comparison methods.
>
> 4)  a) To comprehensively assess the GPT-4/GPT-4V model on TextureDesigner, we utilize 25 test prompts, organized into 5 distinct thematic categories. The ToT-Selection results are listed in **Appendix D**. The "World Peace" and "Colorful World" themes are more open-ended compared to "Future Technology." GPT-4V exhibits more diverse responses on the open-ended themes, suggesting a better understanding of human intent. However, for the more constrained "Future Technology" theme, GPT-4V is more consistent than GPT-4. b) Comparisons are illustrated in Figure 19. GPT-4V exhibits a more accurate and nuanced ability to describe visual content compared to LLava. As demonstrated in our initial experiment, LLava occasionally produces "hallucinations" in its lighting descriptions, particularly when evaluating consistency with the given prompt. In contrast, GPT-4V accurately identifies discrepancies between the described lighting conditions and the actual visual content. Consequently, GPT-4V often assigns lower scores to images compared to LLava.
>
> 5)  a)  Refer to Reviewer \#3 Q3. b) The phenomenon where "Renaissance" is generated as part of the text is attributed to the LoRA model, which may have been trained on a dataset containing images with the word `Renaissance.' This occurrence is not a universal phenomenon in our experiments (for more examples, see Figure 13), and it certainly highlights the importance of higher-quality LoRA models.
>
> [1] Word-As-Image for Semantic Typography. ACM Trans. Graph. 42(4): 151:1-151:11 (2023)
>
> [2] DS-Fusion: Artistic Typography via Discriminated and Stylized Diffusion. ICCV 2023: 374-384
>
> [3] WordArt Designer: User-Driven Artistic Typography Synthesis using Large Language Models. EMNLP 2023: 223-232

---

> > ### Comment · Reviewer_zqVY · 2024-11-26
> >
> > 1. The visualization of the outputs of each step in the full pipeline still appears to be missing (Questions.1). Such visualization is important for the accuracy of the described pipeline in the paper. As mentioned in Questions.1, the subfigures in Fig. 10  (like the picture in the lower left corner) may be good choices for illustrating these intermediate steps, since those are the "final results".
> >
> > 2. The focus of Questions.2 is on how the LoRAs were collected, the parameter count for each LoRA, the data requirements for training each LoRA, and other relevant details. These details are still missing.
> >
> > 3. For the response to Question 5, as the  Reviewer9F4p's concern suggests, in my view, creativity and readability are not inherently conflicting in the design. It is possible for both to reach a high standard. Therefore, "balance" does not seem to be an accurate description.

---

> ### Author Response · Authors · 2024-11-27
> **Response to the additional feedback**
>
> Thank you for your additional feedback!
>
> 1) Please refer to the Figure 20 for the details of each step. The Glyph image generated by the "GlyphDesigner" is illustrated in Figure 23, and the effect of the different fonts for glyph transformation is illustrated in Figure 22.
>
> 2) **Model Information and Access**: We've compiled information about the models used in this project into a file (model_info.json) accessible at [Anonymous Link](https://shorturl.at/Hdl37).  These models leverage the power of the open-source community; they were downloaded from [Civitai](https://civitai.com/) and [Libib](https://www.liblib.art/), where you'll find detailed introductions by their authors.  For our purposes, we've categorized the models by style (see Figure 11) and selected them based on Tree-of-Though (ToT).
>
>      **Model Characteristics**: LoRA models typically have 10+ million to 100+ million parameters, depending on the LoRA rank (typically 32-128).  A small portion of our library includes standard Stable Diffusion (SD) base models.  Thanks to safetensors, switching between base models is very fast (around 0.5 seconds).  LoRA models themselves typically require dozens to hundreds of images for training.  Community-developed base models are often derived by combining and improving existing models, rather than starting from scratch.
>
> 3) We fully concur with your perspective: creativity and readability are not inherently conflicting in the design. We would like to offer a slight clarification regarding our response to Reviewer 9F4p Q3, where we intended to convey that the integration of text with the background and the readability of the text itself requires a balance. That is to say, when we aim for a more harmonious integration of text with the background, we may sacrifice some readability, and conversely, when we enhance the readability of the text, the degree of integration with the background may decrease. We apologize for the initial imprecise use of the term `creativity'.
>
>
>     Regarding the enhancement of readability, we have conducted some explorations, but we have found that there is a significant inaccuracy in the automated assessment of the readability of artistic fonts. Even the most optimal Optical Character Recognition (OCR) methods currently available, such as PP-OCRv4, have an extremely low recognition rate for artistic fonts (10%<), whereas humans can effectively identify artistic fonts by integrating contextual information.  Consequently, we are unable to construct an automated pipeline to accurately perceive the objective impact of variations in glyph shapes and the extent of texture rendering on readability.
>
>     At the current stage, achieving precise control over the degree of artistic refinement to ensure that both creativity (artistic quality) and readability are maintained at a high level requires human intervention in judging various combinations of changes. This makes the design process more complex and time-consuming, which goes against the original intention of our system's ease of use. Therefore, how to ensure that the generated results possess a high level of both creativity (artistic quality) and readability is a highly valuable question. The core lies in the high-precision automated perception of artistic quality and readability, which is also the central direction of our future research.
>
> We will continue to refine the manuscript before the deadline, adding more details and examples to improve reader comprehension.

---

### Official Review · Reviewer_t5qA · 2024-11-02

**Soundness:** 3
**Presentation:** 4
**Contribution:** 4
**Rating:** 6
**Confidence:** 4

**Summary:**

This paper introduces a novel framework for artistic typography synthesis that leverages Large Language Models (LLMs) to drive an interactive and user-centered design process. The system is built on a multi-agent architecture, including Pipeline, Glyph, and Texture agents, which enable the generation of highly customizable WordArt.
The authors introduce a comprehensive feedback mechanism that integrates insights from multimodal models and user evaluations to iteratively refine hyperparameters, ensuring that outputs align with specific stylistic and thematic preferences.

**Strengths:**

**Innovative Framework** The use of a multi-agent system (Pipeline Designer,
Glyph Designer, Texture Designer, and Q&A Evaluation Agent) and the integration of large model to drive the design process is a novel approach that addresses the challenges of creating high-quality WordArt.

**User-Centric Design** The comprehensive feedback mechanism, which includes insights from multimodal models and user evaluations, ensures that the generated WordArt aligns closely with user preferences and expectations.

**Multilingual Capabilitie** The system demonstrates robust performance across English, Chinese, Japanese, and Korean, making it versatile and suitable for a wide range of applications.

**Empirical Validation** The paper provides detailed comparisons with state-of-the-art methods, showing that MetaDesigner outperforms existing techniques in terms of WordArt synthesis success, quality, diversity, and creativity.

**Weaknesses:**

**Complexity and Usability** The multi-agent system and the iterative optimization process may be complex for non-technical users, potentially limiting the system's accessibility.

**Computational Resources** The paper does not provide a detailed analysis of the computational resources required to run MetaDesigner, which is important for practical deployment.

**Questions:**

**Glyph and Texture** Will separating Glyph and Texture limit the diversity of generated results? Could you provide additional ablation studies to demonstrate the necessity of decoupling Glyph and Texture?

**User Interface** Can you provide more details about the user interface and how users interact with the system? How intuitive is the user experience?

**Failure Case** It would be best to explain the existing artifacts in the current system and suggest potential areas for improvement in future work.

---

> ### Author Response · Authors · 2024-11-25
> **Response**
>
> ### Response to the Weakness:
> 1)  We appreciate the reviewer's insightful comment regarding the potential complexity of our multi-agent system and iterative optimization process for non-technical users. We agree that user-friendliness is crucial. Therefore, We have developed a user-friendly interface that abstracts away the underlying complexity of the system. Users can interact with the system through simple prompts and intuitive controls, without needing to understand the technical details.
>
> 2) We have listed efficiency, please refer to Review #2-Q3,  all the experiments are conducted on a Tesla V 100 GPU.
>
> ### Response to the Questions：
> 1）We have incorporated additional comparison by separating Glyphs and Texture into the current version of the manuscript, please refer to Figure 18.
>
> 2）Please refer to Figure 21.
>
> 3）Our system does have some failure cases. a) Some images contain redundant text content. As pointed out by Reviewer \#2 (zqVY) with the example of `Renaissance' (Figure 8, lower right), this is due to the low quality of the LoRA model used by Texture Designer, which also highlights the importance of enhancing the quality of the LoRA models employed by Texture Designer. b) The readability of text in some images is insufficient (Figure 5, the last two examples in the 'Ours' row may exhibit this issue). To enhance the text's and background's harmony, we utilize a high control scale, while acknowledging that this may marginally impact readability in some instances. This highlights the need for automated systems capable of quantitatively evaluating both artistic and textual qualities in generated outputs. Our subsequent research will be directed towards developing such systems.

---

> > ### Comment · Reviewer_t5qA · 2024-11-27
> >
> > Thanks for your response.  After reviewing all the author's responses and the revised paper, I am maintaining this positive score.

---

> ### Author Response · Authors · 2024-11-27
> **Response**
>
> Thank you for your additional feedback!  We will continue to refine the manuscript before the deadline, adding more details and examples to improve reader comprehension.

---

### Author Response · Authors · 2024-11-26
**Update Manuscript**

We have updated Table 1 and Table 2 to reflect the impact of evaluation feedback. Additionally, we have included a new Table 3 to present VLM evaluation results. Figure 17 provides further visualizations, while Figure 18 showcases the performance of mixing/separating glyphs and textures. We will continue to refine the manuscript before the deadline, adding more details and examples to improve reader comprehension.

---

### Author Response · Authors · 2024-11-28
**Dear reviewers**

Dear reviewers，

We sincerely appreciate the diligent efforts of all reviewers. We have carefully considered all feedback and made appropriate revisions to the manuscript during the previous discussion period. As the discussion period has been extended for an additional six days, we are eager to continue the dialogue to foster a deeper understanding of our work and further enhance its quality.

---

### Meta-Review · Area_Chair_sCGb · 2024-12-21

**Metareview:**

This paper introduces a multi-agent framework for artistic typography synthesis, consisting of a Pipeline Designer, a Glyph Designer, and a Texture Designer.

The major strengths of the paper: 1) The proposed framework is well motivated and designed, with multi agents and LLM and advanced prompting.  2) It can generate artistic typograph with higher visual quality, text representation, and style, as well as better alignment to user preference, and multi lingual user cases. 3) The whole amount work/contribution is non-trival, and some promising results are shown.

The major weaknesses of the paper: 1) As reviewers pointed out, this is a system work, so the technical contribution itself may not be very high. And the system is quite complicated. 2) Some technical and experiment details are missing or not clear.

**Additional Comments On Reviewer Discussion:**

Reviewer t5qA mainly asked about Usability of the system (e.g. to non-tech users),  computation resources. Reviewer zqVY asked questions about novelty and experiment design, and asked for clarification of several technical/experimental choices. Reviewer 9F4p asked about some missing technical details, questionable experiment designs and unsatisfactory results.

The authors' rebuttal addressed most of the issues, and convinced reviewers to raise their scores.

---

### Decision · Program_Chairs · 2025-01-22

Accept (Poster)